# Asymmetric subgenomic chromatin architecture impacts on gene expression in resynthesized and natural allopolyploid *Brassica napus*

Zeyu Li[1,3], Mengdi Li [1,2,3] & Jianbo Wang [1✉]

Although asymmetric subgenomic epigenetic modification and gene expression have been revealed in the successful establishment of allopolyploids, the changes in chromatin accessibility and their relationship with epigenetic modifications and gene expression are poorly understood. Here, we synthetically analyzed chromatin accessibility, four epigenetic modifications and gene expression in natural allopolyploid *Brassica napus*, resynthesized allopolyploid *B. napus*, and diploid progenitors (*B. rapa* and *B. oleracea*). "Chromatin accessibility shock" occurred in both allopolyploidization and natural evolutionary processes, and genic accessible chromatin regions (ACRs) increased after allopolyploidization. ACRs associated with H3K27me3 modifications were more accessible than those with H3K27ac or H3K4me3. Although overall chromatin accessibility may be defined by H3K27me3, the enrichment of H3K4me3 and H3K27ac and depletion of DNA methylation around transcriptional start sites up-regulated gene expression. Moreover, we found that subgenome $C_n$ exhibited higher chromatin accessibility than $A_n$, which depended on the higher chromatin accessibility of $C_n$-unique genes but not homologous genes.

[1] State Key Laboratory of Hybrid Rice, College of Life Sciences, Wuhan University, Wuhan 430072 Hubei, China. [2] Key Laboratory of Resource Biology and Biotechnology in Western China, Ministry of Education, College of Life Sciences, Northwest University, Xi'an 710069, China. [3] These authors contributed equally: Zeyu Li, Mengdi Li. ✉email: jbwang@whu.edu.cn

Polyploidization events have been found in the evolutionary history in more than 70% of angiosperms[1], driving the evolution and speciation of polyploids with more favorable physiological characteristics and phenotypes than their diploid progenitors[2]. Polyploidization brings two or more sets of identical or distinct genomes into a nucleus, producing either autopolyploids or allopolyploids[3], which is accompanied by dramatic and dynamic genetic and epigenetic changes[3–5], and allopolyploids have long been thought to play a more critical role in plant divergence due to unique hybridization during their formation[4]. Allopolyploids undergo more rapid and drastic transcriptional changes due to the sudden appearance of diverged subgenomes in the same cells[6]. Genomic expression dominance has been demonstrated to be a typical feature of many allopolyploids[7,8], and a large amount of data has demonstrated that genomic expression dominance is associated with asymmetric epigenetic features, such as DNA methylation[8,9], histone modification[8] and chromatin compactness[10]. However, how chromatin accessibility shapes transcriptome patterns during allopolyploidization and subsequent evolutionary processes remains poorly understood.

Regulation of the transcription of genes is governed by interactions between regulatory proteins and cis-regulatory elements (CREs)[11,12]. Active CREs are generally located in accessible chromatin regions (ACRs) that can be accessed by nucleases through evicting or unraveling nucleosomes[13]. Identification of ACRs helps decipher CREs in the genome, which is critical for understanding the complex transcriptional regulatory networks underlying gene expression. The accessibility of chromatin can be assayed by several established methods, including DNase-Seq based on DNase I[14], MNase-Seq based on micrococcal nuclease[12,15], and ATAC-Seq based on transposase Tn5[16,17]. Compared with other methods, ATAC-Seq has some advantages, such as significantly less experimental materials and easier library construction[18]. This method has been applied in plant genomic studies, such as delineating ACRs of multiple plant species[19,20], revealing cell type-specific transcriptional regulatory networks[20–22], and identifying altered chromatin accessibility in mutant plants[18] and plants under stress[23]. However, there are scarce studies on how chromatin accessibility changes after polyploidization.

The chromatin regulatory landscape is related to histone modifications and DNA methylation[24,25]. As one of the most characterized chromatin modifications, histone acetylation is associated with the open chromatin structure and active transcription[26]. Because lysine can be monomethylated, demethylated or trimethylated, histone methylation has more complex forms and functions[27]. Repressive histone methylation (e.g., H3K9me2, H3K9me3 and H3K27me3) is located in heterochromatin and inhibits gene expression[28], whereas active histone methylation (e.g., H3K4me1 and H3K4me3) is located in euchromatin and promotes gene expression[29]. Surprisingly, several studies revealed bivalent chromatin with active (H3K4me3) and repressive (H3K27me3) histone methylation[30,31], and these modifications may represent a more accessible chromatin environment that promotes the binding of regulatory proteins to regulate gene expression[30]. As a conserved epigenetic modification, DNA methylation can be located in the promoter or gene body and play profound roles in gene expression[32–34]. A recent study found that DNA methylation in CG, CHG and CHH contents impacts the chromatin accessibility of heterochromatin[25]. The relationships among histone modifications, DNA methylation and chromatin accessibility during the formation and evolutionary process of allopolyploids are worth exploring.

Brassica napus (2n = 4x = 38, AACC), an allopolyploid plant, is one of the most important oil crops widely cultivated in the world[8,35]. Natural B. napus originated from interspecific hybridization and subsequent whole genome duplication (WGD) of B. rapa (2n = 2x = 20, AA) and B. oleracea (2n = 2x = 18, CC) approximately 7500 years ago in the Mediterranean region[35]. With the whole genomes of these three species sequenced, they became a model system for studying allopolyploidization[35–37]. Although the asymmetric gene distribution[35], asymmetric epigenetic modification[8,38] and bias gene expression[8,38,39] of subgenomes of B. napus have been revealed gradually, the differences in chromatin accessibility between subgenomes and the evolutionary changes of ACRs have not been reported in the allopolyploid B. napus. In this study, we generated genome-wide chromatin accessibility of natural B. napus, resynthesized B. napus and in silico 'hybrid' constructed by two progenitors. Genotype-specific motifs and transcription factor (TF) regulatory networks were illuminated. We comprehensively analyzed differences in chromatin accessibility between subgenomes and further compared the chromatin accessibility of biased expressed homeologous gene pairs. In addition, we comprehensively analyzed the chromatin accessibility and its relationships with three histone epigenetic modifications (H3K4me3, H3K27ac and H3K27me3), DNA methylation and gene expression of natural B. napus, resynthesized B. napus and in silico 'hybrid' constructed by two progenitors.

## Results

**Genome-wide accessible chromatin profiling in B. napus and its progenitors.** To ascertain the chromatin regulatory landscape of two types of B. napus and their progenitors, we constructed genome-wide maps of ACRs of leaves through ATAC-Seq, with three biological replicates for all the genotypes. High Spearman correlations were observed between biological replicates (Supplementary Fig. 1). Our results showed that approximately 64 million (B. rapa, A2) to 102 million (B. oleracea, C2) raw reads were obtained from ATAC-Seq, more than 96% of which were clean reads in all samples (Table S1). According to previous studies[38,40], we constructed an in silico 'hybrid' (A_C), which can fuse well with parental data for the convenience of subsequent studies, by mixing the reads of B. rapa and B. oleracea in equal proportions. As shown in Table S1, an average of approximately 94.7%, 95.6%, 93.0%, and 98.0% of clean reads from the ATAC-Seq data of B. rapa, B. oleracea, resynthesized B. napus, and natural B. napus were mapped to the genome of B. napus[35], and the GC content of each sample was higher than 40%.

In this study, the peaks identified by MACS2[41] were referred to as accessible chromatin regions (ACRs). In total, 36,161, 40,913, and 27,965 ACRs were identified in in silico 'hybrid', resynthesized B. napus and natural B. napus (Fig. 1a), which were associated with 26,601, 29,476, and 21,182 genes (Fig. 1b), respectively. In addition, we defined 45.64% of ACRs as genic ACRs (gACRs, overlapping with genic region from transcriptional start site, TSS, to transcriptional terminate site, TTS), and the remainder as intergenic ACRs (iACRs, not overlapping with genic region) in in silico 'hybrid' (Fig. 1c). Compared with the in silico 'hybrid', the proportion of gACRs increased, whereas the proportion of iACRs decreased in resynthesized B. napus. The proportion of gACRs in natural B. napus was similar to that in resynthesized B. napus. The genes associated with ACRs in the three genotypes were involved in the GO terms 'positive regulation of molecular function', 'positive regulation of protein metabolic process', and 'positive regulation of catalytic activity' (Supplementary Fig. 2, Supplementary Data 1). At chromosomal level, we found that many regions with high chromatin accessibility were conserved among the three genotypes, but there were also many variations in accessible chromatin regions

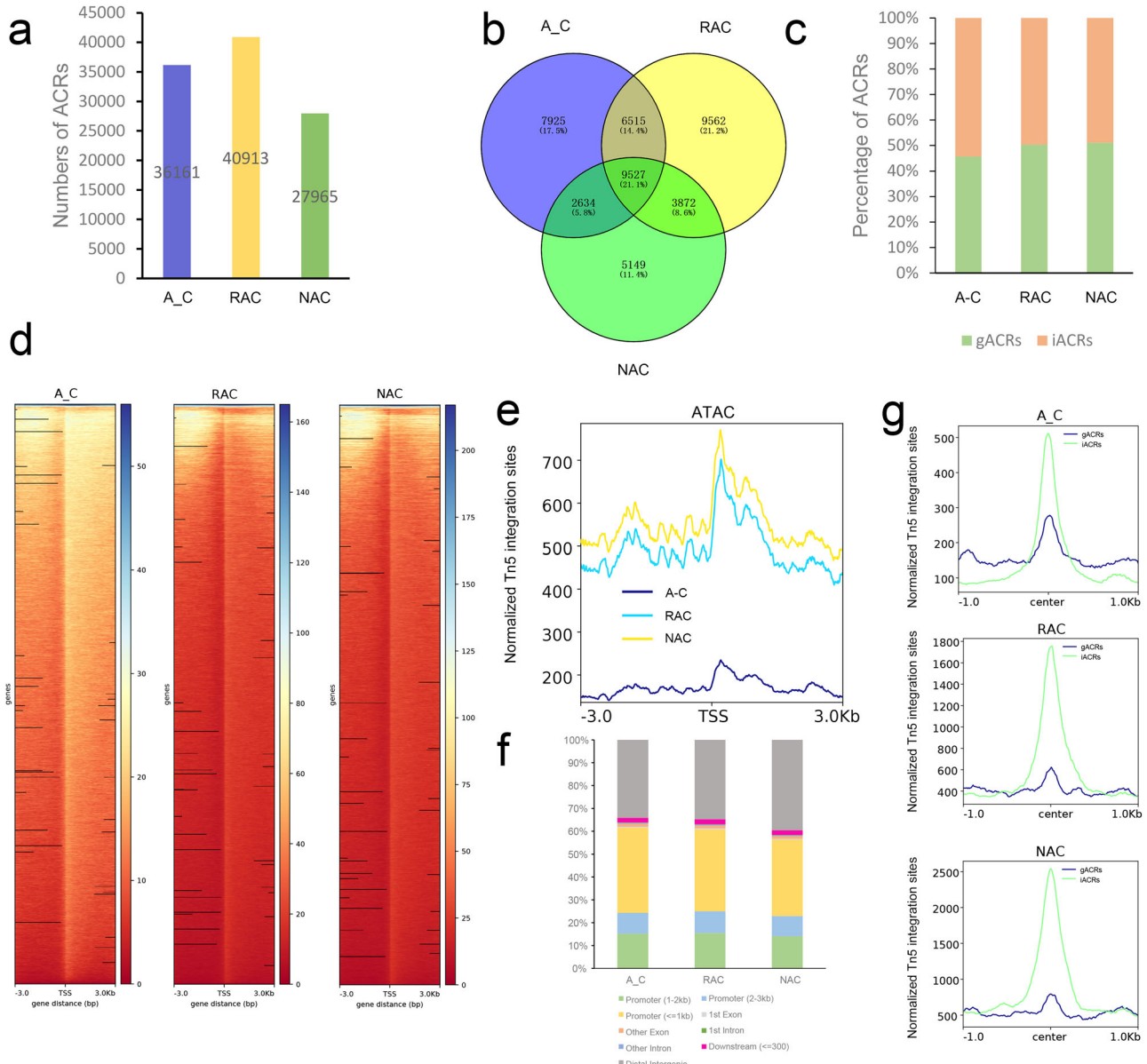

**Fig. 1 Profiling of chromatin accessibility in three genotypes of *B. napus*. a** Numbers of ACRs. **b** Venn diagram of genes associated with ACRs. **c** Percentages of genic and intergenic ACRs. **d** Heatmaps of ACRs. **e** Profiles of ACRs. T test revealed statistical significance of the differences of chromatin accessibility among the three genotypes. **f** Distribution of ACRs in the genome. **g** Profiles of genic and intergenic ACRs. A_C, in silico 'hybrid'; RAC, resynthesized *B. napus*; NAC, natural *B. napus*; ACRs, accessible chromatin regions; gACRs, genic ACRs; iACRs, intergenic ACRs; TSS, transcript start site; center, peak center of ACRs.

or openness among the three genotypes (Supplementary Fig. 3). To explore the distribution of ACRs among genes, we inspected the ATAC-Seq signal at ACRs in three genotypes using deepTools[42]. These analyses revealed that ACRs tended to be highly enriched around the transcript start site (TSS) in all genotypes (Fig. 1d). Surprisingly, in silico 'hybrid' had an average maximum of approximately 235 reads per kilobase per million mapped reads (RPKM) around TSS, whereas resynthesized *B. napus* and natural *B. napus* had higher accessibility in these regions, with an average maximum of approximately 700 and 770 RPKM around TSS, respectively (Fig. 1e). These results implied that the overall chromatin accessibility increased in resynthesized *B. napus* during polyploidization coupled with the hybridization process, and the overall chromatin accessibility also increased in natural *B. napus* during the subsequent evolution process. Then, we mapped these ACRs to genomic features, and found that the

majority (61.55%) of ACRs were distributed in the promoter region, especially promoters within 1 kb (37.24%), followed by the distal intergenic region (34.09%), whereas only 2.3% and 2.06% of ACRs were distributed in the gene body (exons and introns) and within 300 bp downstream of gene end in the in silico 'hybrid' (Fig. 1f). Similar proportions of these ACRs were observed in resynthesized *B. napus*. Compared with resynthesized *B. napus*, the proportion of ACRs distributed in the promoter region decreased 4.45%, and the proportion of ACRs distributed in the distal intergenic region increased 4.76%, whereas the proportions of ACRs distributed in the gene body and within 300 bp downstream of the gene end were almost unchanged in natural *B. napus* (Fig. 1f). These results implied the far more important regulatory roles of ACRs in the promoter region and distal intergenic region than in the gene body and within 300 bp downstream of the gene end. Although the distribution of these

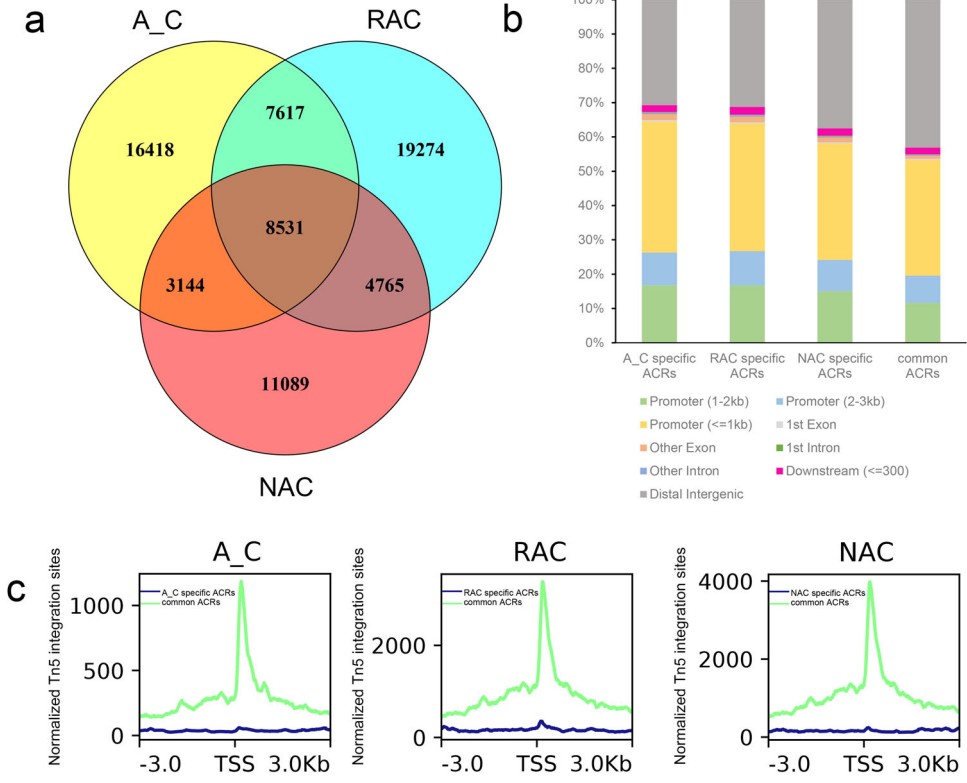

**Fig. 2 Chromatin accessibility differences among the three genotypes. a** Overlap of ACRs in in silico 'hybrid', resynthesized *B. napus* and natural *B. napus*. **b** Distribution of ACRs in in silico 'hybrid', resynthesized *B. napus* and natural *B. napus*. **c** Profiles of common and genotype-specific ACRs. A_C, in silico 'hybrid'; RAC, resynthesized *B. napus*; NAC, natural *B. napus*; ACRs, accessible chromatin regions; center, peak center of ACRs.

ACRs hardly changed during allopolyploidization, the ACRs in distal intergenic regions seemed to play an increasingly important role in natural evolutionary processes. In addition, we compared the ATAC-Seq signals of two types of ACRs and found that iACRs had higher ATAC-Seq signal levels around the peak center than gACRs in all genotypes (Fig. 1g), which may imply that the ACRs in genic regions need less chromatin accessibility to regulate gene expression.

**Chromatin accessibility changed dramatically in resynthesized and natural allopolyploid *B. napus*.** To examine the changes in chromatin accessibility during the allopolyploidization process, we overlapped the ACRs identified in the three genotypes. We found that genotype-specific ACRs were more abundant than common ACRs (Fig. 2a). A total of 8531 ACRs were common among three genotypes, whereas 16,418, 19,274, and 11,089 genotype-specific ACRs were identified in in silico 'hybrid', resynthesized *B. napus* and natural *B. napus*, respectively. The abundant genotype-specific ACRs indicated drastic changes in chromatin accessibility, both during the formation and natural evolution process of *B. napus*. Then, we compared the distribution of the two types of ACRs in the genomic region and found that the distribution difference was mainly in the promoter region and distal intergenic region, although the sum of these two regions was almost unchanged (Fig. 2b). More common ACRs were distributed in the distal intergenic region (5.6−12.4%) but less of that distributed in the promoter region (4.8−11.2%) than genotype-specific ACRs (Fig. 2b). As showed in Fig. 2c, common ACRs were more enriched than genotype-specific ACRs especially near the peak center, indicating that common ACRs were more open than genotype-specific ACRs.

Since many ACRs were identified as common ACRs, we wondered if there were quantitative differences in common ACRs

between genotypes. Only those ACRs that had a fold change of 2 or more ($P$ value < 0.05) in a genotype were categorized as different enriched ACRs (DEAs) in that genotype and we identified 320 genes that had more accessible DEAs and 500 genes that had fewer accessible DEAs in in silico 'hybrid' than in resynthesized *B. napus* (Supplementary Fig. 4). The results of GO analysis showed that resynthesized *B. napus*-enriched DEAs-related genes were involved in 'cell wall organization' and 'lipid metabolic process', whereas in silico 'hybrid'-enriched DEAs-related genes were involved in 'chromosome organization' and 'organelle organization' (Supplementary Data 2). A total of 556 genes had different accessible ACRs between the two types of *B. napus*, in which 415 DEA-related genes were enriched in natural *B. napus* and 141 DEA-related genes were enriched in resynthesized *B. napus*. Natural *B. napus*-enriched DEAs-related genes were associated with some metabolic processes, such as 'ncRNA metabolic process', whereas resynthesized *B. napus*-enriched DEAs-related genes were involved in 'biosynthetic process' and 'gene expression'. There were 592 genes that had natural *B. napus*-enriched DEAs and 200 in silico 'hybrid'-enriched DEAs, and these genes were involved in multiple biological processes, such as 'mRNA process' and 'photosynthesis'. These results indicated that genes exhibiting different ACRs played important roles in metabolic processes and gene expression regulation.

**The effects of chromatin accessibility on gene expression in *B. napus* and its progenitors.** To determine the relationship between chromatin accessibility and gene expression levels, we explored the chromatin accessibility of four groups of genes divided by gene expression levels: highly expressed genes (high; transcripts per million reads (TPM) greater than 10), moderately expressed genes (med; TPM between 1 and 10), lowly expressed

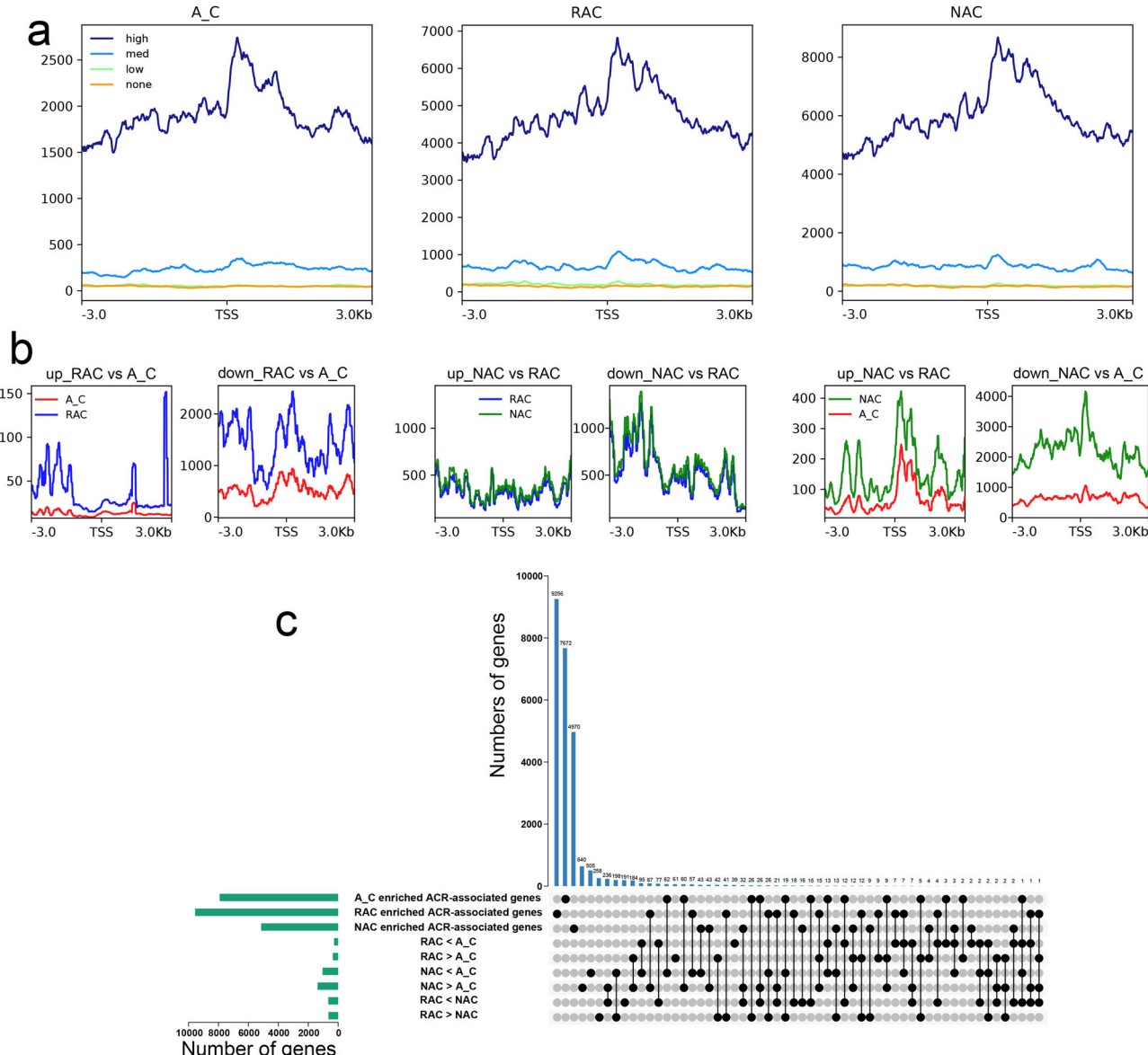

**Fig. 3 The relationship between chromatin accessibility and gene expression level. a** Chromatin accessibility of genes with different expression levels. High, highly expressed genes (TPM greater than 10); med, moderately expressed genes (TPM between 1 and 10); low, lowly expressed genes (TPM greater than 0 and less than 1); none, silenced genes (TPM equal to 0). **b** Chromatin accessibility of DEGs. DEGs, genes with |log$_2$ fold change | ≥1 and adjusted *P* value ≤ 0.001. **c** Overlap of genotype-enriched ACR-associated genes and DEGs. Genotype-enriched ACR-associated genes, genes related to genotype-specific ACRs and ACRs with ACR that had a fold change ≥2. A_C, in silico 'hybrid'; RAC, resynthesized *B. napus*; NAC, natural *B. napus*; TSS, transcript start site; ACRs, accessible chromatin regions.

genes (low; TPM greater than 0 and less than 1), and silenced genes (none; TPM equal to 0). As shown in Fig. 3a, highly expressed genes had the highest ATAC-Seq signal, followed by moderately expressed genes. The ATAC-Seq signal of lowly expressed genes and silenced genes was the lowest. These results indicated that chromatin accessibility positively affected gene expression. Then, we wondered whether differential gene expression was caused by changes in chromatin accessibility. The distribution of expression levels of differentially expressed genes (DEGs) were shown in Supplementary Fig. 5. Surprisingly, compared with the in silico 'hybrid', both up-regulated and down-regulated DEGs had higher ATAC-Seq signals in resynthesized and natural *B. napus* (Fig. 3b), which may be caused by the higher whole ATAC-Seq signals of the two types of *B. napus*. Up- and down-regulated DEGs had higher ATAC-Seq signals in natural *B. napus* than in resynthesized *B. napus*, which was

consistent with the higher whole ATAC-Seq signal in natural *B. napus*. Interestingly, the ATAC-Seq signals of down-regulated DEGs was higher than that of up-regulated DEGs in all comparisons. These results indicated that the changes in chromatin accessibility between genotypes did not seem to explain the differential gene expression due to differences in overall chromatin accessibility between genotypes and that the down-regulated DEGs needed higher chromatin accessibility to maintain gene transcription. To examine how many DEGs could be regulated by ACRs, we overlapped the genotype-enriched ACR-associated genes (genes related to genotype-specific ACRs and genotype-enriched DEAs) and DEGs. In each comparison, the highly expressed DEGs in a genotype were referred to as genotype-enriched DEGs. As shown in Fig. 3c, 17.4% (46) of in silico 'hybrid'-enriched DEGs and 17.3% (53) of resynthesized *B. napus*-enriched DEGs were associated with genotype-enriched

ACRs. In the comparison of two types of *B. napus*, 15.5% (101) of the resynthesized *B. napus*-enriched DEGs and 14.3% (95) of the natural *B. napus*-enriched DEGs were regulated by ACRs. Although most DEGs were found in comparison of in silico 'hybrid' and natural *B. napus*, the proportion of genotype-enriched ACR-associated DEGs was similar to that in the previous comparisons. These results indicated that the changes in chromatin accessibility regulated the differential expression of a part of DEGs, which was similar in maize[22].

**Identification of *cis*- and *trans*-regulators in ACRs of *B. napus* and its progenitors**. Since active *cis*-regulatory elements (CREs) are generally embedded in ACRs that can be integrated by *trans*-regulatory proteins to regulate gene expression, we identified over-represented motifs from ACRs of each genotype. In total, 82, 84, and 71 motifs and corresponding transcription factor (TF) families (Supplementary Data 3) were identified and 21, 22, and 18 of them were unique in in silico 'hybrid', resynthesized *B. napus* and natural *B. napus*, respectively (Supplementary Fig. 6a). Then, we detected 23,907, 28,602, and 19,855 genes that were target genes of these TFs in in silico 'hybrid', resynthesized *B. napus* and natural *B. napus*, respectively. We selected the top 4 TFs that regulated the most DEGs and visualized their regulatory network with DEGs (Supplementary Fig. 6b–d). We overlapped these target genes and DEGs, and found 37.5%, 34.6%, and 35.6% DEGs were regulated by these TFs in comparisons of in silico 'hybrid' *vs*. resynthesized *B. napus*, in silico 'hybrid' *vs*. natural *B. napus* and resynthesized *B. napus vs*. natural *B. napus*, respectively (Supplementary Fig. 6e). These results implied that many genotype-unique over-represented motifs were bound by corresponding TFs to regulate the differential expression of genes.

**Local epigenetic modifications affected chromatin accessibility in *B. napus* and its progenitors**. To examine the relationship between chromatin accessibility and local epigenetic modifications, we compared three histone modification statuses and DNA methylation levels around the ACRs. Surprisingly, not only active histone modification (H3K4me3 and H3K27ac) but also repressive histone modification (H3K27me3) were found to be highly enriched around the peak center of ACRs (Supplementary Fig. 7), whereas DNA methylation except CHG content was deficient around the peak center of ACRs (Supplementary Fig. 8). To determine which histone modification had the greatest impact on ACR openness, we divided ACR into six clusters: cluster 1 was associated with H3K27me3, cluster 2 was associated with H3K4me3, cluster 3 was associated with H3K27ac, cluster 4 was associated with H3K27me3 and H3K4me3, cluster 5 was associated with H3K4me3 and H3K27ac, and cluster 6 was associated with H3K27me3 and H3K27ac. We found that cluster 1 had the highest ATAC-Seq signal, followed by cluster 3 in the in silico 'hybrid' (Fig. 4a, d). Cluster 6 had the highest ATAC-Seq signal, followed by cluster 1 in resynthesized *B. napus* (Fig. 4b, d), whereas cluster 1 had the highest ATAC-Seq signal, followed by cluster 6 in natural *B. napus* (Fig. 4c, d). These results indicated that H3K27me3 had the greatest impact on ACR openness in the in silico 'hybrid', whereas the bivalent histone modifications H3K27me3 and H3K27ac had the greatest impact on ACR openness in resynthesized *B. napus*, and the influence of H3K27me3 gradually became dominant in natural *B. napus*.

To further determine the relationship between ACR intensity and local histone modifications, we divided the ACRs into 10 groups by ranking their pileup from low to high for each genotype. All groups of ACRs exhibited higher three histone modification signals than both flanks (Fig. 5a–c), whereas all groups except rank 9 or rank 10 exhibited lower CG, CHG, and

CHH DNA methylation levels than both flanks (Supplementary Fig. 9). Surprisingly, the intensity of ACRs was positively associated not only with three histone modification levels, but also with the DNA methylation level (Supplementary Fig. 10). Additionally, we compared four local epigenetic modifications around gACRs and iACRs and found that iACRs exhibited the much higher H3K27ac and H3K27me3 and DNA methylation levels than gACRs, whereas iACRs showed higher H3K4me3 around the peak center but lower H3K4me3 in both flanks than gACRs (Fig. 5e, f and Supplementary Fig. 9e, f). Since H3K27me3 has long been considered an inhibitory histone modification, we wondered why it was positively correlated with chromatin accessibility. Therefore, we examined the relationship between histone modification level and gene expression and found that highly expressed genes not only had the highest H3K4me3 and H3K27ac, but also had the highest H3K27me3 signal, followed by moderately expressed genes (Supplementary Fig. 11). Similar to the ATAC-Seq signals, the H3K27me3 signals of lowly expressed genes and silenced genes were almost absent. Although both lowly expressed genes and silenced genes had H3K4me3 and H3K27ac signals, the signal of the former was much higher than that of the latter. In contrast to the active histone modifications H3K4me3 and H3K27ac, the DNA methylation level was found to be negatively associated with the gene expression level (Supplementary Fig. 12).

Since the overall chromatin accessibility did not account for differential gene expression, we examined the histone modification level of up- and down-regulated DEGs. Similar to the ATAC-Seq signals, down-regulated DEGs showed higher H3K27me3 levels than up-regulated DEGs except in comparison of resynthesized *B. napus* and natural *B. napus* (Fig. 6c). Up-regulated DEGs exhibited lower H3K4me3, H3K27ac, and H3K27me3 levels, whereas down-regulated DEGs showed higher H3K4me3, H3K27ac, and H3K27me3 levels in in silico 'hybrid' than that in two types of *B. napus* (Fig. 6a–c). However, in the comparison of resynthesized *B. napus* and natural *B. napus*, up-regulated DEGs exhibited higher H3K4me3, H3K27ac, and H3K27me3 levels, whereas down-regulated DEGs showed lower H3K4me3, H3K27ac, but higher H3K27me3 levels in resynthesized *B. napus*. In contrast to active histone modification, up-regulated DEGs had lower DNA methylation levels, whereas down-regulated DEGs had higher DNA methylation levels (Supplementary Fig. 13). These results suggested that the chromatin accessibility of DEGs was correlated with the H3K27me3 level. The up-regulation of DEGs seemed to require active histone modifications (H3K4me3 and H3K27ac), but lower DNA methylation levels in the three genotypes.

**Asymmetrical chromatin accessibility and epigenomic modifications between subgenomes**. *B. napus* contains two distinct subgenomes derived from *B. rapa* and *B. oleracea*[35]. To examine which subgenome is more dominant, we initially compared gene expression levels of two subgenomes in leaves of three genotypes, and found that gene expression levels of subgenome $C_n$ were significantly higher than subgenome $A_n$ (Supplementary Fig. 14a). To compare the chromatin accessibility between two subgenomes, we counted the number of ACRs in two subgenomes and found that ACRs were more in all genomic regions of the $C_n$ subgenome (Fig. 7a). There was no significant difference between intensity of ACRs between two subgenomes of in silico 'hybrid', whereas the intensity of ACRs of subgenome $A_n$ was significantly higher than that of $C_n$ of natural *B. napus* and resynthesized *B. napus* (Fig. 7b). In in silico 'hybrid', the intensity of intergenic ACRs in subgenome $A_n$ was significantly higher than that of $C_n$, but the intensity of genic (including exon, intron,

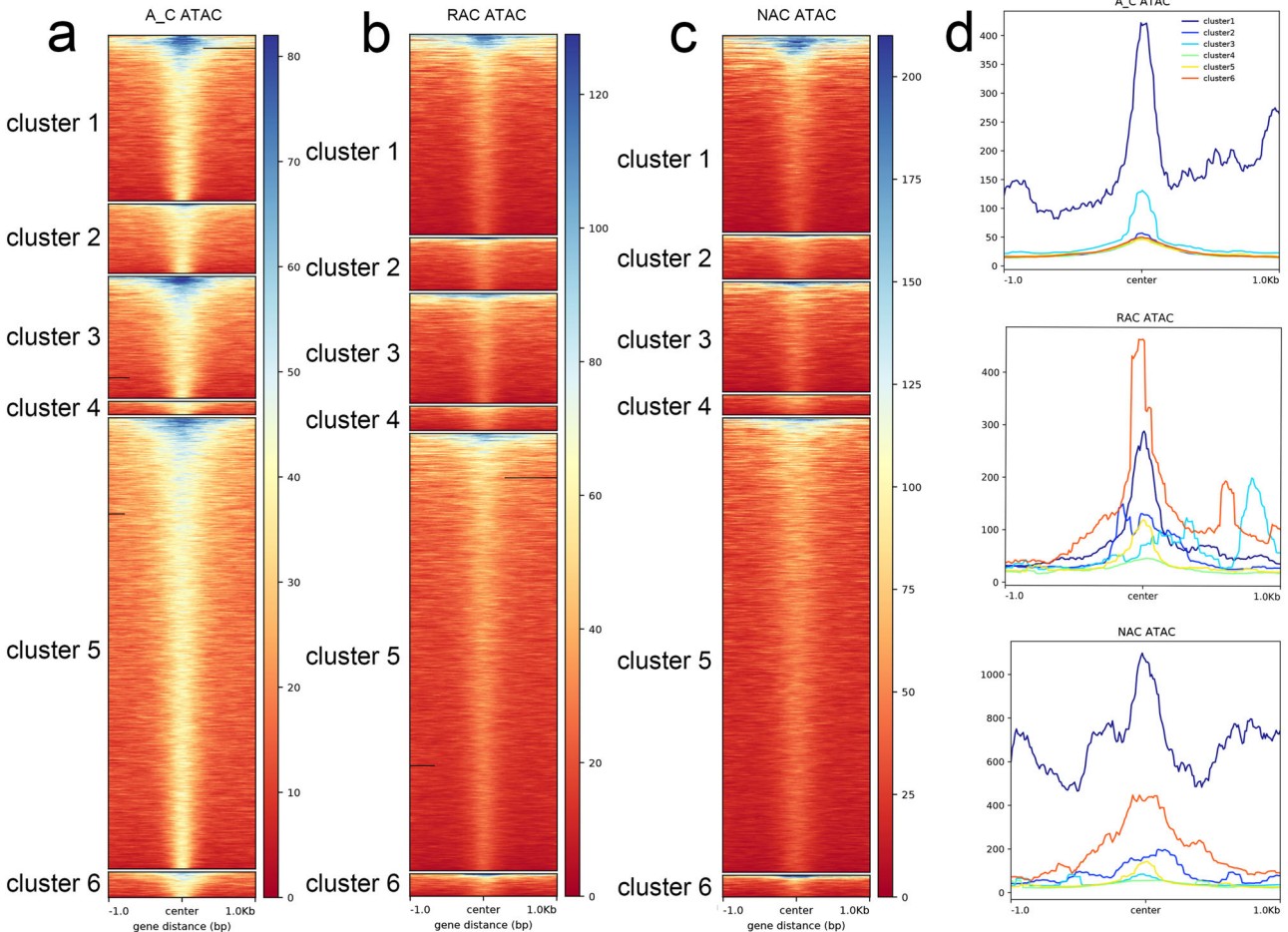

**Fig. 4 Clustering analysis of histone modification-associated ACRs. a** Heatmaps of 6 clusters of ACRs in the in silico 'hybrid'. **a** Heatmaps of 6 clusters of ACRs in the in silico 'hybrid'. **b** Heatmaps of 6 clusters of ACRs in resynthesized *B. napus*. **c** Heatmaps of 6 clusters of ACRs in natural *B. napus*. **d** Profiles of 6 clusters of ACRs in three genotypes. A_C, in silico 'hybrid'; RAC, resynthesized *B. napus*; NAC, natural *B. napus*; ACRs, accessible chromatin regions; center, peak center of ACRs.

promoter-TSS and TTS) ACRs was significantly lower than that of $C_n$ (Fig. 7c). The intensity of intergenic ACRs in subgenome $A_n$ was also significantly higher than that of $C_n$ in natural *B. napus* and resynthesized *B. napus*, but the differences of intensity of genic ACRs were attenuated in resynthesized *B. napus* and not even significant in natural *B. napus*. Further analysis revealed that intergenic ACRs showed higher H3K4me3, H3K27ac, H3K27me3 modifications and DNA methylation than genic ACRs (Supplementary Figs. 15 and 16). Then, we compared the chromatin accessibility, H3K4me3, H3K27ac, H3K27me3 modifications and DNA methylation between two subgenomes. Genes in subgenome $C_n$ had higher average ATAC-Seq signals, H3K27me3 levels, and DNA methylation levels than the genes in subgenome $A_n$, whereas genes in subgenome $C_n$ had slightly lower average H3K4me3 and H3K27ac levels than genes in subgenome $A_n$ (Supplementary Figs. 14b–d and 17). To further explore the cause of the differences between the two subgenomes, we divided the genes of each genotype into homeologous genes and subgenome-unique genes according to our previous study[38]. As shown in Fig. 8a, subgenome $C_n$-unique genes exhibited the highest expression level in each genotype. Subgenome $C_n$-unique genes had significantly higher gene expression than those in subgenome $A_n$-unique in each genotype, whereas there was no significant difference between homeologous genes in two subgenomes except natural *B. napus*. Then, we investigated the epigenetic modifications of these genes, and found that

subgenome $C_n$-unique genes showed the highest ATAC-Seq signal and H3K27me3 level in all genotypes (Fig. 8b, e). Surprisingly, homeologous genes had higher H3K4me3 and H3K27ac levels but lower DNA methylation levels than subgenome unique genes in each subgenome in all genotypes (Fig. 8c, d; Supplementary Fig. 18), which indicated that homeologous gene pairs may need more H3K4me3 and H3K27ac to finely regulate gene expression. Taken together, these results suggested that the higher overall gene expression of subgenome $C_n$ may be due to the high expression of subgenome $C_n$-unique genes, which had higher chromatin accessibility and H3K27me3 levels.

According to our previous study[38], homeologous gene pairs were divided into A-/C-biased expression genes (A-/C-BEGs) and no biased expression genes (no-BEGs). The distribution of gene expression levels of homeologous gene pairs is shown in Fig. 9a. Although a similar ATAC-Seq signal and H3K27me3 were observed between genes in two subgenomes of A-/C-BEGs and no-BEGs, we found that $A_n$ genes had higher H3K4me3 and H3K27ac levels between 100 bp upstream and downstream of TSS in A-BEGs, whereas $C_n$ genes had higher H3K4me3 and H3K27ac levels between 100 bp upstream and downstream of TSS in C-BEGs, and $A_n$ genes and $C_n$ genes had similar H3K4me3 and H3K27ac levels in no-BEGs in all genotypes (Fig. 9b–e). In contrast to the distribution of active histone modifications, genes with higher expression had lower DNA methylation level than their homeologous genes between 200 upstream and 700 bp

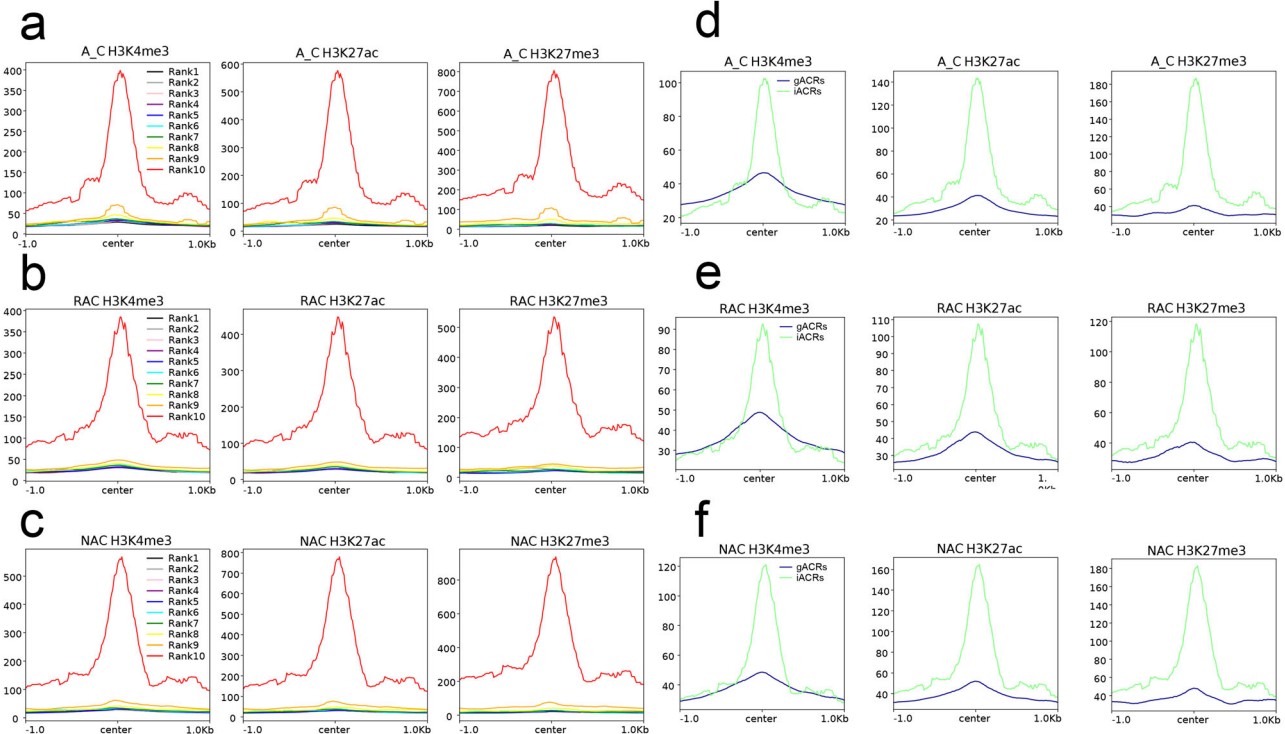

**Fig. 5 Histone modifications around ACRs. a** H3K4me3, H3K27ac, and H3K27me3 modifications around ACRs with different chromatin accessibility in A_C. **b** H3K4me3, H3K27ac, and H3K27me3 modifications around ACRs with different chromatin accessibility in RAC. **c** H3K4me3, H3K27ac, and H3K27me3 modifications around ACRs with different chromatin accessibility in NAC. Values of 1 to 10 denote groups with low to high chromatin accessibility, respectively. **d** H3K4me3, H3K27ac, and H3K27me3 modifications around gACRs and iACRs in A_C. **e** H3K4me3, H3K27ac, and H3K27me3 modifications of gACRs and iACRs in RAC. **f** H3K4me3, H3K27ac, and H3K27me3 modifications of gACRs and iACRs in NAC. A_C, in silico 'hybrid'; RAC, resynthesized *B. napus*; NAC, natural *B. napus*; ACRs, accessible chromatin regions; gACRs, genic ACRs; iACRs, intergenic ACRs; center, peak center of ACRs.

downstream of TSS, and no BEGs had similar DNA methylation levels in the two subgenomes (Supplementary Fig. 19). Surprisingly, genes in subgenome $C_n$ had higher DNA methylation levels than their homeologous genes in $A_n$ on both sides of TSS regardless of gene expression bias. These results indicated that homeologous gene pairs had more H3K4me3 and H3K27ac modifications to finely regulate biased gene expression, which was seemingly independent of chromatin accessibility, and the gene expression was positively associated with H3K4me3 and H3K27ac levels but negatively related to DNA methylation levels around TSS.

**Chromatin accessibility of singleton and five types of duplicated genes in *B. napus*.** Allopolyploid *B. napus* is rich in duplicated genes due to multiple round WGDs during the evolution process[35]. According to a previous study[43], we divided genes in *B. napus* into six types of genes: singletons (genes that have one copy), WGD-derived genes (genes derived from WGD), TD-derived genes (genes derived from tandem duplication), TRD-derived genes (genes derived from transposed duplication), DSD-derived genes (genes derived from dispersed duplication), and PD-derived genes (genes derived from proximal duplication). The TD-, TRD-, DSD-, and PD- derived genes were referred to as small-scale duplicated genes[43]. As shown in Fig. 10a, singletons had significantly higher gene expression than all duplicated genes in the three genotypes. Among duplicated genes, WGD-derived genes had the highest gene expression levels, followed by DSD- and TRD-derived genes, and PD- and TD-derived genes had the lowest gene expression levels in all genotypes. We wondered whether chromatin accessibility was related to the regulation of

gene expression of these types of genes, and detected the ATAC-seq signal of these genes. Consistent with the higher gene expression level, singletons had higher chromatin accessibility than duplicated genes in all genotypes (Fig. 10b). WGD-derived genes had higher chromatin accessibility than small-scale duplicated genes, which may be related to the higher gene expression of WGD-derived genes. However, unlike the gene expression levels, TRD- and PD-derived genes had higher ATAC-Seq signals than DSD- and TD-derived genes. These results indicated that singletons and WGD-derived genes were more accessible than small-scale duplicated genes and were coupled with higher gene expression levels.

## Discussion

Allopolyploids have long been thought to be more likely to establish and have a greater contribution to plant divergence due to WGD coupled with interspecific hybridization[4]. Newly formed allopolyploids need to reorganize distinct genomes from different parental species[44], which disrupts genetic and epigenomic features[3–5] and results in altered DNA methylation[8,9,38], histone markers[8,38], chromatin compactness[10], sRNA production[45], and gene expression[7,8,38]. These whole genome-wide changes contribute to the survival and successful establishment of newly formed allopolyploids[46], which are related to novel phenotypic variation[47,48]. Investigating the consequences of allopolyploidization across both recent and deep time scales is more conducive to studying the evolutionary processes of polyploidy. In this study, we selected resynthesized *B. napus*, natural *B. napus* and their progenitors *B. rapa* and *B. oleracea* to explore the evolutionary processes across two different time scales (the new

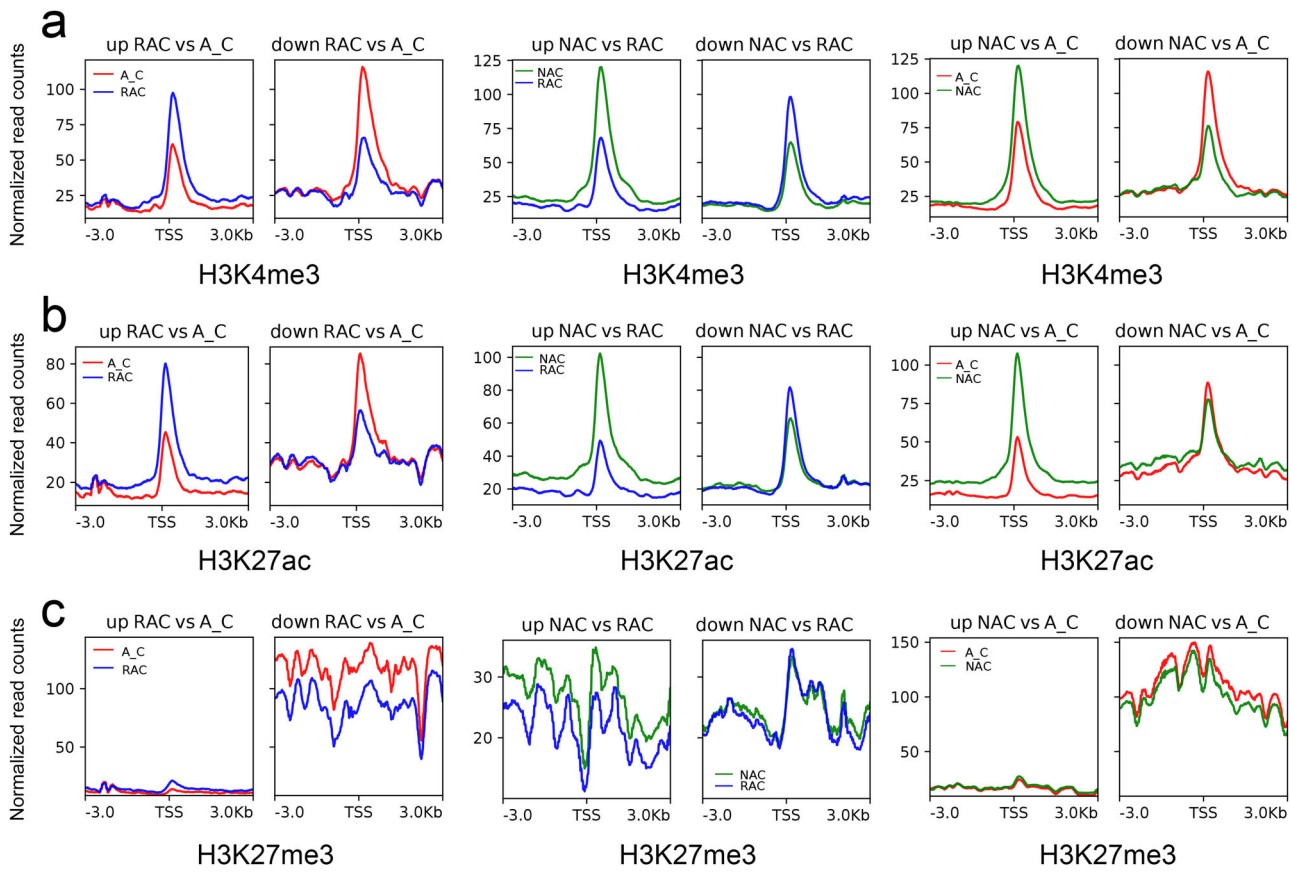

**Fig. 6 The relationship between DEGs and three histone modifications. a** The relationship between DEGs and H3K4me3. **b** The relationship between DEGs and H3K27ac. **c** The relationship between DEGs and H3K27me3. DEGs, differentially expressed genes; A_C, in silico 'hybrid'; RAC, resynthesized *B. napus*; NAC, natural *B. napus*; TSS, transcriptional start site.

formation and natural establishment experienced approximately 7500 years). The genome-wide chromatin accessibility and the potential relationship between chromatin accessibility and gene expression during the new formation and natural evolution process of *B. napus* were comprehensively analyzed.

During the successful evolutionary process, allopolyploids must face "genomic shocks", which result in dramatic gene expression alterations[44,49–51]. However, there are few studies on the changes in chromatin accessibility and its relationship with gene expression during allopolyploidization. With advances in technologies, genome-wide chromatin accessibility regions can be obtained. In this study, we used the assay for transposase-accessible using sequencing (ATAC-Seq) to detect genome-wide ACRs in natural *B. napus*, resynthesized *B. napus* and in silico 'hybrid' constructed with their progenitors. A total of 27,965 to 40,913 ACRs were identified in the three genotypes, which is comparable to the number of ACRs in several plant species[13,52]. Surprisingly, the ATAC-Seq signal of ACRs in resynthesized *B. napus* was significantly higher than that in in silico 'hybrid' but was significantly lower than that in natural *B. napus*, which indicated that both allopolyploidization and subsequent evolution increased chromatin accessibility of *B. napus*. The gACRs increased, whereas iACRs decreased in both synthesized *B. napus* and natural *B. napus* compared with the in silico 'hybrid'. These observations indicated that allopolyploid *B. napus* with more complex genomes may require closer *cis*-regulatory elements embedded in gACRs to regulate gene transcription than diploid progenitors. Unlike the majority of identified common ACRs in different cell types of Arabidopsis[21], more than half of ACRs were genotype-specific ACRs in our study (Fig. 2a). This suggested that

allopolyploid *B. napus* was subjected to tremendous changes in chromatin accessibility, which we defined as "chromatin accessibility shock" during allopolyploidization. Interestingly, the common ACRs showed far more accessibility than genotype-specific ACRs (Fig. 2c). Functional annotation results showed that genes related to these common ACRs were involved in the regulation of gene expression and metabolic processes, whereas genes related to genotype-specific ACRs were involved in lipid and steroid biosynthetic processes (Supplementary Data 4). These observations strongly indicated that the common ACRs, which played important roles in gene expression regulation, remained highly accessible in the three genotypes. The low chromatin accessibility of genotype-specific ACRs, which were involved in the metabolic process of organic substances, may be beneficial for rapidly increasing chromatin accessibility and regulating gene expression in response to drastic genome changes.

Due to the great difference among the average ATAC-Seq signals around the TSS of the three genotypes, we did not find genotype-preferentially enriched DEGs with higher chromatin accessibility (Fig. 3b). However, chromatin accessibility was found to be positively associated with overall gene expression in each genotype (Fig. 5a), which was consistent with other plant species previously published[13,22]. We found that approximately a quarter of the DEGs were related to genotype-enriched ACRs in all comparisons (Fig. 3c), similar to the proportion of DEGs that might be affected by chromatin accessibility in different tissues of maize[22]. Furthermore, approximately 35% of DEGs may be regulated by genotype-unique TFs in *B. napus* (Supplementary Fig. 6c). These observations implied the important regulatory roles of chromatin accessibility in the gene expression of different plants.

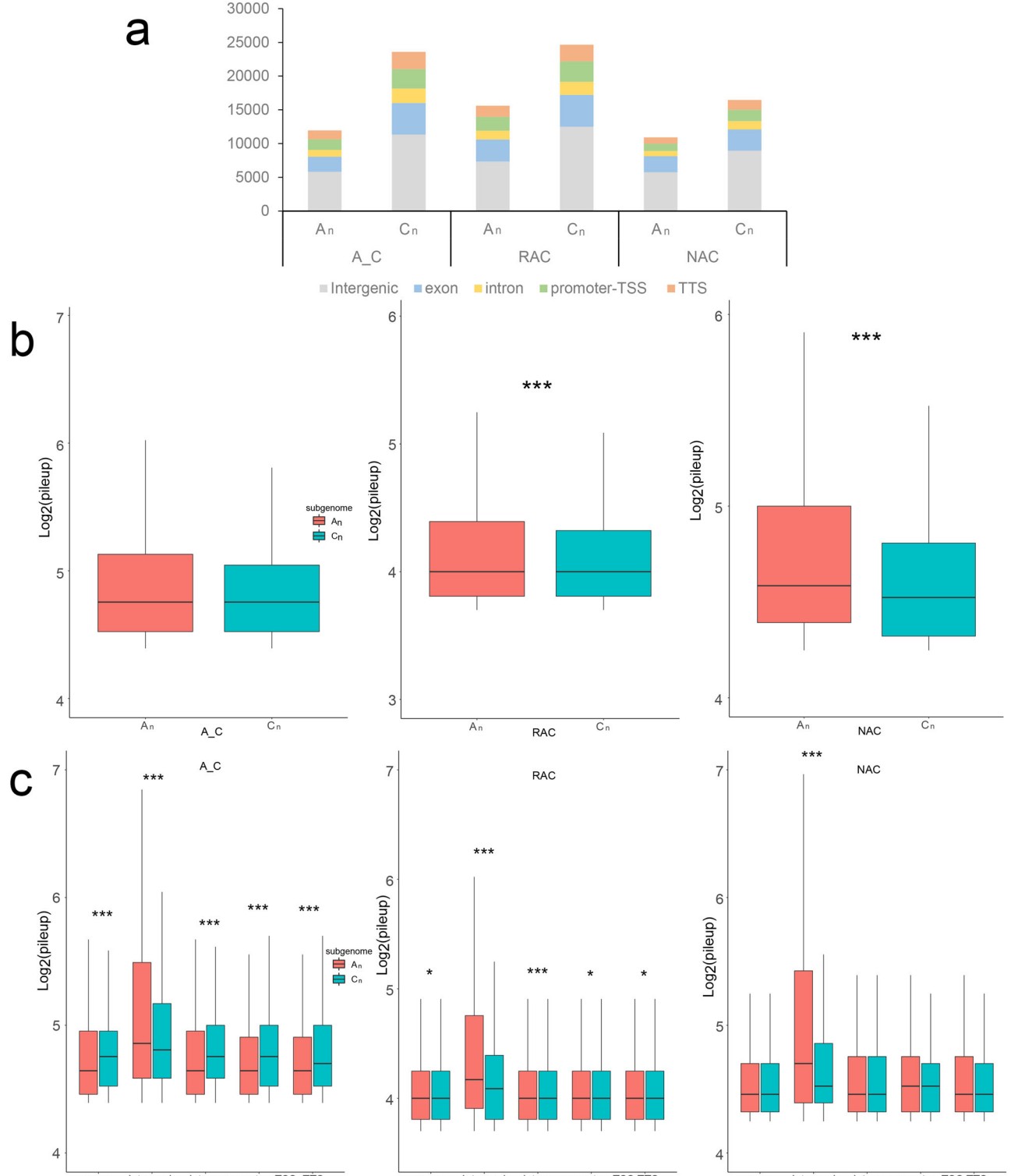

**Fig. 7 Comparison of the chromatin accessibility between two subgenomes. a** Number statistics of ACRs in genomic regions of two subgenomes.
**b** Comparison of the intensity of ACRs between two subgenomes. **c** Comparison of the intensity of ACRs in genomic regions between two subgenomes.
A_C, in silico 'hybrid'; RAC, resynthesized *B. napus*; NAC, natural *B. napus*; pileup, the intensity of ACR. Statistical analysis was conducted using Wilcoxon
rank sum test. *$p < 0.05$ and ***$p < 0.001$.

The local epigenetic modifications of ACR may affect its accessibility, which may affect the differential expression of genes[22,24]. Histone acetylation is generally associated with the open chromatin structure[26], whereas histone methylation can define open or closed chromatin states depending on the degree of methylation and position of the amino acid residue[53]. In our study, the intensity of ACRs was positively correlated not only with active histone modifications, H3K4me3 and H3K27ac, but also with repressive histone modification H3K27me3 (Fig. 5a). In addition, all three epigenetic modifications were found to have

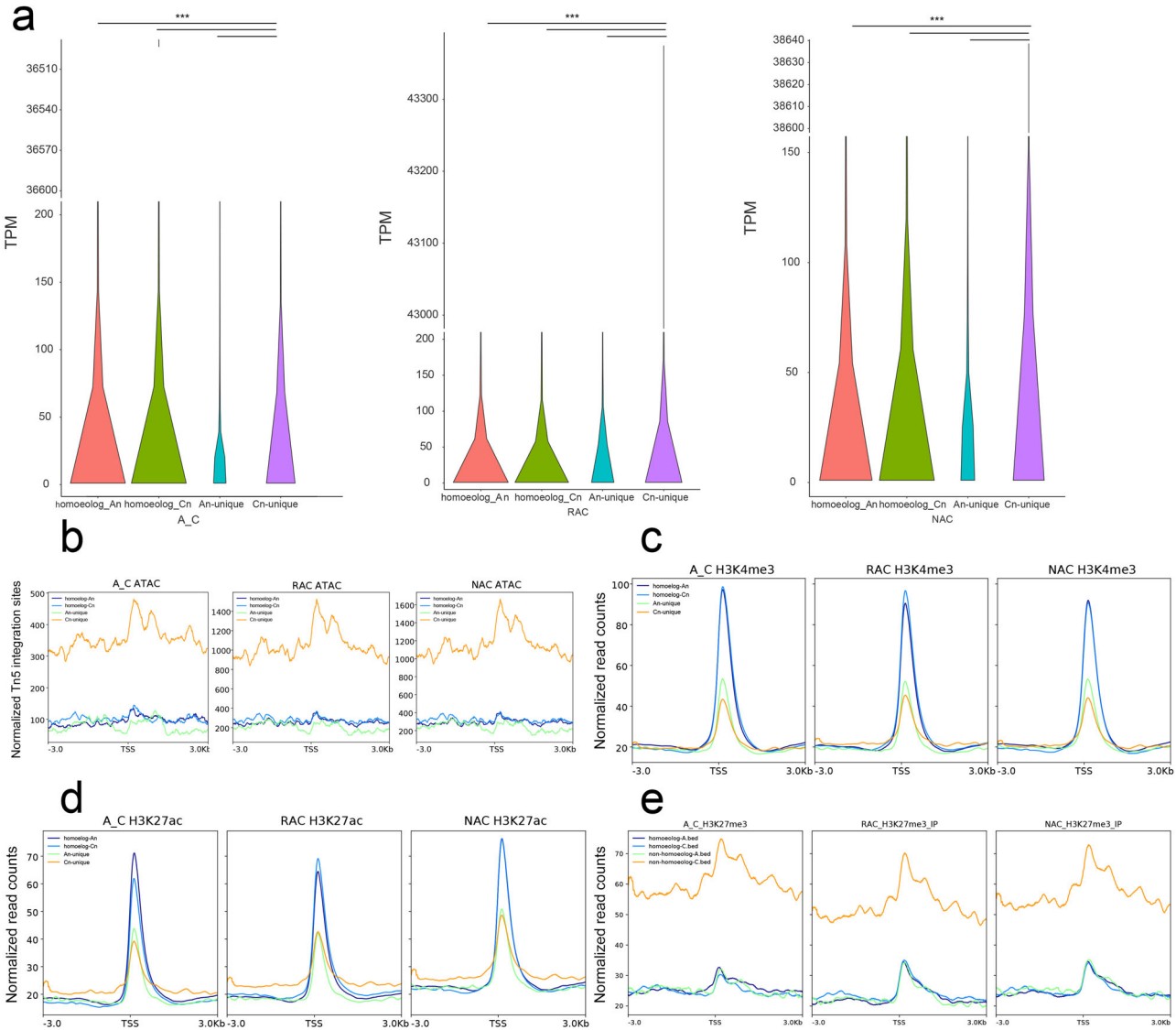

**Fig. 8 Comparison of gene expression levels and four epigenetic modifications among homeologous genes and subgenome-unique genes in two subgenomes. a** Gene expression levels of homeologous genes and subgenome-unique genes in the two subgenomes. **b** Chromatin accessibility of homeologous genes and subgenome-unique genes in two subgenomes. **c** H3K4me3 level of homeologous genes and subgenome-unique genes in two subgenomes. **d** H3K27ac level of homeologous genes and subgenome-unique genes in two subgenomes. **e** H3K27me3 level of homeologous genes and subgenome-unique genes in two subgenomes. A_C, in silico 'hybrid'; RAC, resynthesized *B. napus*; NAC, natural *B. napus*; TSS, transcriptional start site.

positive effects on gene expression (Supplementary Fig. 7). H3K27me3 was found to have a greater influence on ACR openness than H3K4me3 and H3K27ac, and the bivalent histone modifications H3K27me3 and H3K27ac play important roles in chromatin openness during the allopolyploidization process (Fig. 4d). H3K27me3 has long been thought to be a repressive modification that is associated with gene silencing[53–55], and it is found to be enriched in silenced genes with lower chromatin accessibility[13,56]. However, a previous study found that active genes showed significantly higher levels of H3K27me3, whereas silenced genes displayed lower levels of H3K27me3 in cold-treated tubers of potato[30]. In our study, highly and moderately expressed genes with higher ATAC-Seq signals exhibited higher H3K27me3 than lowly expressed and silenced genes (Fig. 3a and Supplementary Fig. 10) in the three genotypes. These observations reflected the complex roles of H3K27me3 in defining chromatin status and gene expression regulation among different plant species. Interestingly, lowly expressed genes had similar accessibility and H3K27me3 modification levels to silenced genes.

In addition, although the H3K4me3 and H3K27ac modification levels of lowly expressed genes were lower than those of highly and moderately expressed genes, they were much higher than those of silenced genes (Supplementary Fig. 11). These results indicated that the prerequisite for transcriptional activation of lowly expressed genes was not accessible chromatin, but active histone modifications (H3K4me3 and H3K27ac) in *B. napus*. DNA methylation was found to be depleted in ACRs in many plants[13,22]. The depletion of DNA methylation was observed in the two edges of ACRs in *B. napus* and its progenitors (Supplementary Fig. 8). Although DNA methylation levels were found to be positively related to the intensity of ACRs (Supplementary Fig. 9), DNA methylation indeed negatively affected gene expression in *B. napus* and its progenitors (Supplementary Fig. 12). Increased chromatin accessibility influenced by DNA methylation[25] and loss of Polycomb Group complexes[57] was not inevitably accompanied by increased gene expression. These observations indicated that the effect of local DNA methylation on chromatin accessibility was separate from the effect of

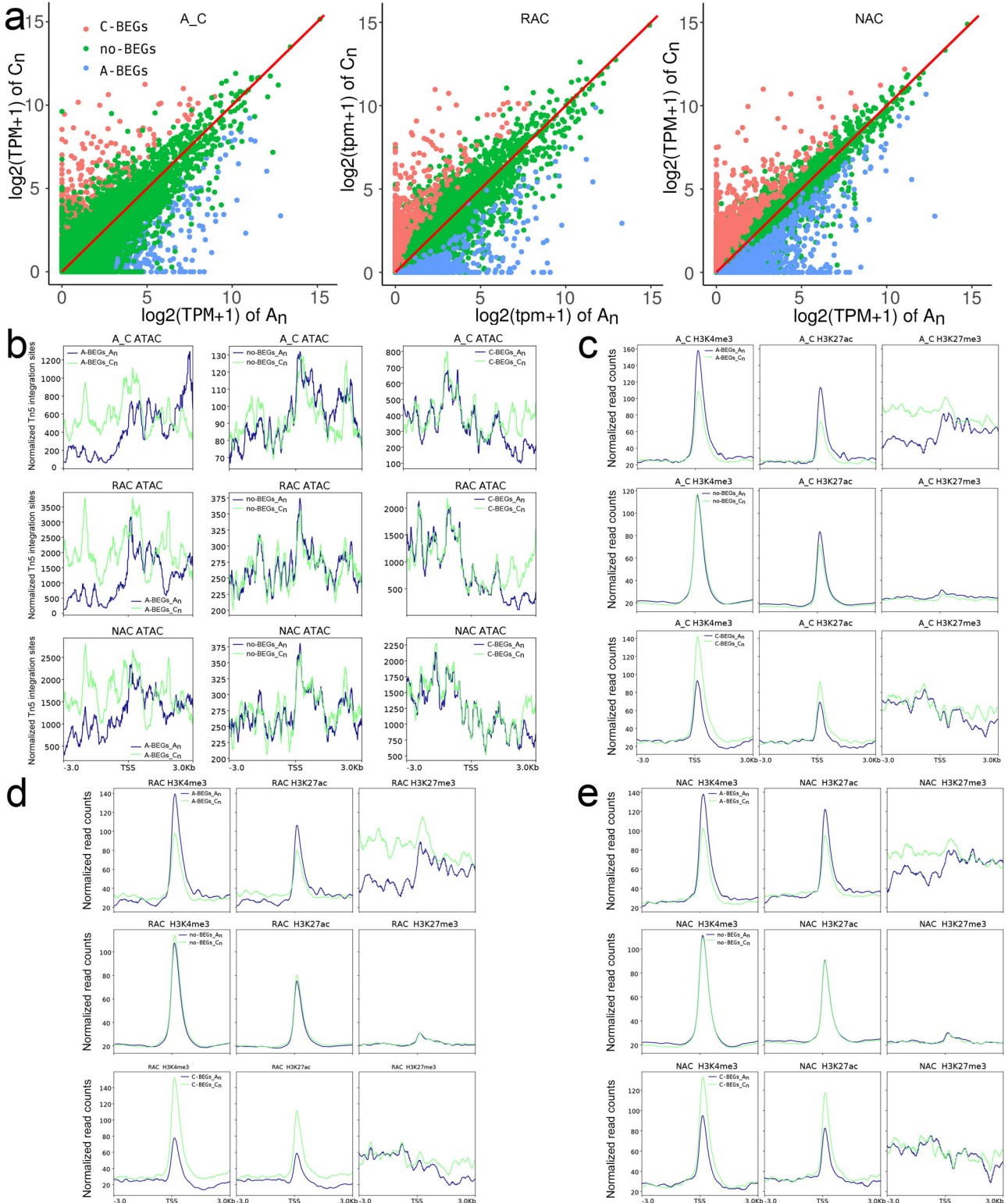

**Fig. 9 Comparison of gene expression levels and five epigenetic modifications between homeologous gene pairs. a** The distribution of gene expression levels of homeologous gene pairs. **b** Chromatin accessibility of homeologous gene pairs. **c** H3K4me3, H3K27ac and H3K27me3 levels of homeologous gene pairs in A_C. **d** H3K4me3, H3K27ac and H3K27me3 levels of homeologous gene pairs in RAC. **e** H3K4me3, H3K27ac and H3K27me3 levels of homeologous gene pairs in NAC. A_C, in silico 'hybrid'; RAC, resynthesized *B. napus*; NAC, natural *B. napus*; TSS, transcriptional start site.

chromatin accessibility on gene expression, and accessible chromatin did not necessarily activate transcription.

The allopolyploids were challenged by reorganizing divergent genetic material from different parental species during formation[44]. The "genomic shock", large-scale conflict of

distinct subgenomes, often results in genome-wide expression dominant of one subgenome, which is referred to as subgenome dominance[58]. The phenomenon that unbalanced epigenetic modifications can regulate biased gene expression between subgenomes has been observed in many allopolyploids, such as

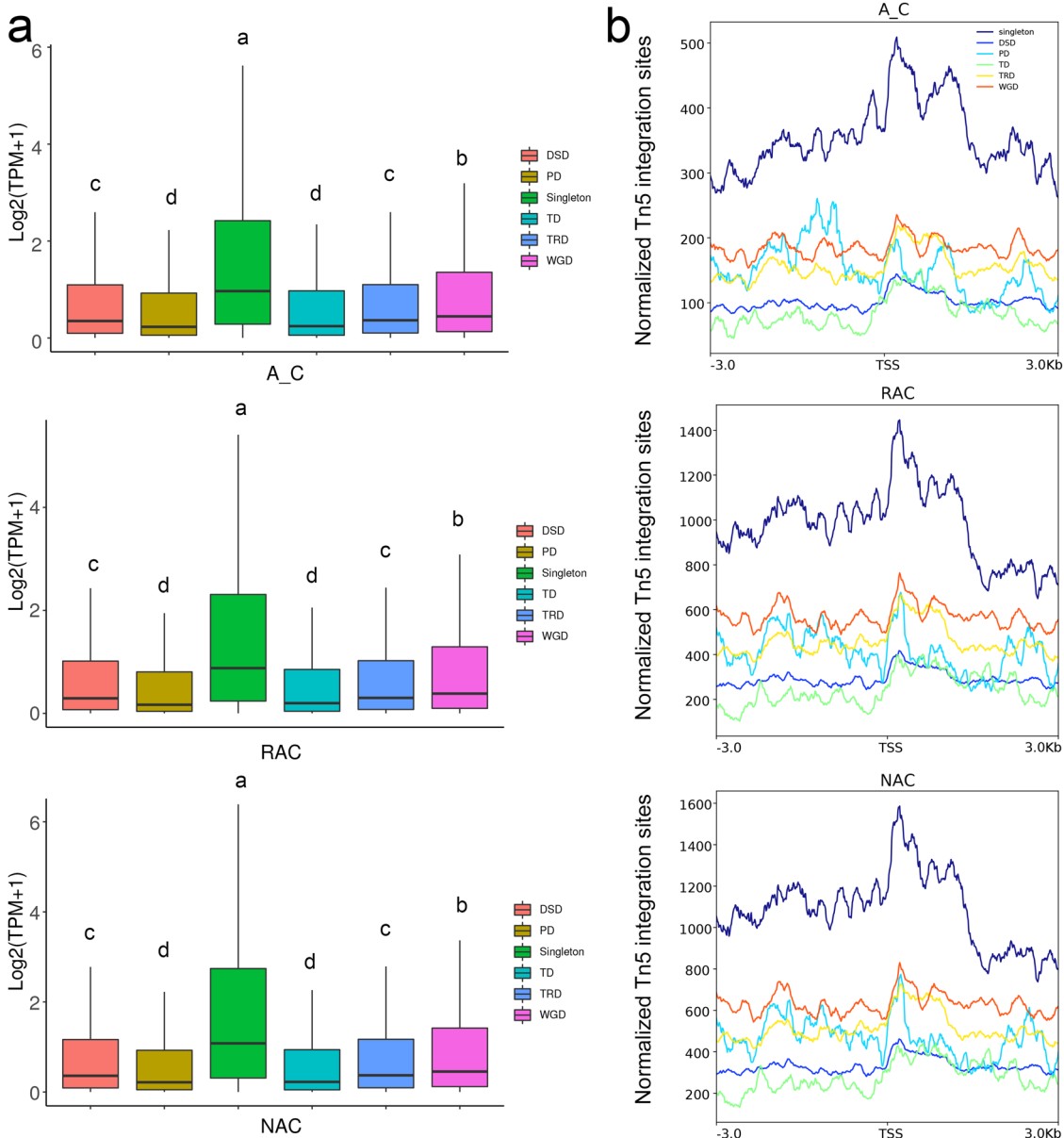

**Fig. 10 Gene expression levels and chromatin accessibility of six types of genes. a** Gene expression levels of six types of genes. **b** Chromatin accessibility of six types of genes. A_C, in silico 'hybrid'; RAC, resynthesized *B. napus*; NAC, natural *B. napus*; TSS, transcriptional start site. DSD, dispersed duplication-derived genes; PD, proximal duplication-derived genes; TD, tandem duplication-derived genes; TRD, transposed duplication-derived genes; WGD, whole-genome duplication-derived genes.

cotton[59], wheat[60] and monkeyflower[9]. In resynthesized *B. napus*, subgenome dominance is partly related to DNA methylation difference between two subgenomes over the first ten generations postpolyploidisation[61]. In maize, DNA methylation and chromatin accessibility drive biased fractionation between two ancient subgenomes[62]. However, the chromatin dynamic following allopolyploidy and its relationship with subgenome expression bias are comparatively poorly explored. In this study, although chromatin accessibility in *B. napus* increased significantly after polyploidization (Fig. 1d, e), subgenome $C_n$ exhibited higher chromatin accessibility than subgenome $A_n$ (Supplementary Fig. 14b). $C_n$ subgenome had more ACRs not only in intergenic regions but also in genic regions (Fig. 7a). We found that the higher accessibility of subgenome $C_n$ was dependent on the higher accessibility of subgenome $C_n$-unique genes (Fig. 8b). The distribution of H3K27me3 but not H3K4me3 or H3K27ac modification was similar to the ATAC-Seq signal in the two subgenomes, so we speculated that the overall chromatin accessibility of genes was determined by H3K27me3.

## Methods

**Plant materials**. Seeds of natural allotetraploid *B. napus* L. (cv. Darmor), resynthesized allotetraploid *B. napus* L. (HC-2) and its parents *B. rapa* L. (cv. 9JC002, paternal), *B. oleracea* L. (cv. 3YS013, maternal), were obtained from the Oil Crop Research Institute, Chinese Academy of Agricultural Sciences, China. The resynthesized *B. napus* was generated by embryo rescue and genome doubling of hybrid of *B. rapa* and *B.oleracea*. Seeds of the four plant materials were germinated on culture dishes with two layers of moist filter paper in light incubators (23 °C, photoperiod is day/night for 16 h/8 h). Young seedlings with fully unfolded cotyledons were transferred into mixed soil (equal amounts of nutritive soil and vermiculite) moistened with 1/2 Hoagland's nutrient solution (pH: 5.8). The transferred seedlings continued to grow in light incubators and were regularly watered with 1/2 Hoagland's nutrient solution (pH: 5.8). Young leaves of five-week-old plants were harvested with three biological replicates and frozen immediately in liquid nitrogen.

**ATAC-Seq and data analysis**. ATAC-Seq from four plant materials was performed according to a previous study[63]. Briefly, plant leaf tissue was ground in liquid nitrogen, and the grinded powder was washed with PBS and treated with lysis buffer. The nuclei were obtained by density gradient centrifugation for 30 min at 120,000 g in 60% Percoll and 2.5 M sucrose solution. Purified nuclei were resuspended in 50 μl transposase integration reaction and then incubated for 30 min at 37 °C. Tagmented DNA was purified by 2× DNA clean beads and then amplified 15 cycles with i5 and i7 index primers (Table S2) to obtain the primary libraries. The primary libraries were purified using 1.5× NGS beads. The purified libraries, including DNA inserts between 50 to 150 bp, were sequenced using a Novaseq 6000 sequencer (Illumina) with the PE150 model.

Raw ATAC-Seq data were first trimmed 3' adaptor, and low-quality reads were removed by Trimmomatic software (v.0.36)[64]. The clean data of each library were further eliminated duplication by FastUniq (v.1.1), and then deduplicated reads were aligned to the reference *B. napus* genome using bowtie2 software (v 2.2.6)[65] with default parameters. Samtools (v.0.1.19)[66] was used to convert the mapped reads in .sam format to .bam format. Peak calling was carried out by MACS2 software to identify ATAC-Seq peaks[41]. ATAC-Seq peaks with fold enrichment greater than 3.4 remained for further analysis, and peaks called in this method were referred to as accessible chromatin regions (ACRs). The quantitative difference in common ACRs was calculated by a Python script, and ACRs with $|\log_2$ fold change$| \geq 1$ and $P$ value < 0.05 were identified differentially enriched ACRs (DEAs). HOMER (v.4.1.0)[67] was used to identify the motifs of ACRs. Bedtools (v.2.25.0)[68] was used for peak annotation. The genomic distribution of ACRs was identified by 'annotatePeak' of ChIPseeker[69]. For visualization of ATAC-Seq data, the bam files from three biological replicates were merged and sorted by Samtools (v 0.1.19)[66]. The sorted bam files were converted to bigWig (bw) files by 'bamcoverage' in deepTools[42]. Heatmaps and average plots were generated by the 'computeMatrix' followed by 'plotHeatmap' and 'plotProfile' in the deepTools 2.0.

**RNA-Seq and data analysis**. Third-generation sequencing libraries were constructed by using the same plant materials used for ATAC-Seq with three biological replicates. The process of constructing RNA-Seq libraries can be found in our previous study[6]. Raw RNA-Seq data were filtered (length > 100, quality >7) by NanoFilt (v.2.5.0) to generate clean data. Pinfish (v1.0) process was used to analyze clean reads. Firstly, full-length reads were identified by Pychopper (https://hpc.nih.gov/apps/pychopper). Secondly, the full-length reads were aligned to the reference *B. napus* genome v.5 (http://www.genoscope.cns.fr/brassicanapus/data/) by Minimap2. Then the clustered alignment results were aligned to the reference genome again by Minimap2 to obtain consensus reads. The gene expression levels were normalized by the transcripts per million reads (TPM). DEGs were identified by the DESeq2 method[70], and the thresholds were $|\log_2$ fold change$| \geq 1$ and adjusted $P$ value (padj) ≤ 0.001. GO analysis for genes was conducted by KOBAS software (v 2.1.1), and the terms with corrected $P$ values ≤ 0.05 were inferred to be significantly enrichment GO terms.

**ChIP-Seq and data analysis**. The ChIP-Seq libraries were constructed by using the same plant materials used for ATAC-Seq with three biological replicates. The process of constructing ChIP-Seq libraries can be found in our previous study[38]. Trimmomatic software (v.0.36) was used to trim the 3' adaptor and remove low-quality reads from the raw data of ChIP-Seq. The clean data eliminated by FastUniq (v.1.1) were aligned to the reference *B. napus* genome using Bowtie2 software (v 2.2.6) with default parameters. The ChIP-Seq peaks were identified by MACS2 software. For visualization of ChIP-Seq data, three bam files were merged and sorted by Samtools (v 0.1.19) and then the sorted bam files were converted to bigWig (bw) files by 'bamcoverage' in deepTools. Heatmaps and average plots were performed by the 'computeMatrix' followed by 'plotHeatmap' and 'plotProfile' in the deepTools 2.0.

**DNA methylation data analysis**. The DNA methylation libraries were constructed by using the same plant materials used for ATAC-Seq with three biological replicates. Genomic DNA was extracted from 12 samples by cetyl trimethyl-lammonium bromide (CTAB) method. The lambda DNA spike-in was used to correct for non-conversion rates of uracil by adding 1 ng of methyl-free lambda DNA to 1 μg genomic DNA as an internal reference for the conversion test. The EZ DNA Methylation Gold Kit (Zymo Research, Irvine, Ca, USA) was used to conduct bisulfite conversion of DNA. The Acegen Bisulfite-Seq Library Prep Kit (AG0311; Acegen, Shenzhen, China) was used to construct whole genome sulfite sequencing (WGBS) libraries and sequenced on Illumina HiSeq X10 (30-fold sequencing depth). FastQC was used to evaluate WGBS data and Trimmomatic (v.0.36) was used to filter raw reads. The clean reads were mapped to the *B. napus* genome using BatMeth2[71]. Batmeth2 calculates regional DNA methylation level by weighting the sequencing depth of a region[72]. The methylation level of each cytosine site is calculated by dividing the methylation reads by the coverage of that site. Locations with coverage of less than 3 were considered missing data. The methylation level of a genomic region is determined by dividing the sum of the methylated reads in that region by the sum of the echo cytosine site coverage.

**Building the in silico 'hybrid'**. The in silico 'hybrid' was constructed by mixing ATAC-Seq, ChIP-Seq, and RNA-Seq data of *B. rapa* and *B. oleracea* at a ratio of 1:1, respectively. Data of DNA methylation were mixed in proportion to the genome size of *B. rapa* and *B. oleracea* to construct in silico 'hybrid'[38]. Three in silico 'hybrid' were constructed according to three biological replicates of all sequencing data in each genotype.

**Statistics and reproducibility**. The statistical significance in our study was determined by R (https://r-project.org). The *wilcox.test* function and *chisq.test* function in the R package were used to implement the Wilcoxon rank sum test and $\chi^2$ test.

**Reporting summary**. Further information on research design is available in the Nature Research Reporting Summary linked to this article.

## Data availability

The data of ATAC-Seq, RNA-Seq, ChIP-Seq, and DNA methylation are available in the National Center for Biotechnology Information (NCBI) Sequence Read Archive (SRA) with the accession numbers SRR17818023-SRR17818034, SRR13302173-SRR13302184, SRR13318007-SRR13318030, and SRR13306925-SRR13306936, respectively. All source data in this study are available in Supplementary Data 1–5.

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

## Acknowledgements
We thank Xiaoming Wu for providing the experimental materials. We thank Dahu Zou, Yi Li, and Jing Jin and Guoliang Li for their valuable advice on data analysis. This work was supported by the National Natural Science Foundation of China (31970241).

## Author contributions
J.W., and Z.L. designed this research. Z.L., and M.L. completed data analysis. Z.L. performed the research and wrote the manuscript. M.L. was responsible for planting materials and providing the data of histone modifications and DNA methylation. J.W. revised the manuscript. All authors read and approved the final manuscript.

## Competing interests
The authors declare no competing interests.

**Additional information**

