## [Peer Review File · Communications Biology]

Reviewers' comments:

Reviewer #1 (Remarks to the Author):

Li and colleagues present an impressive body of work interrogating differences in chromatin accessibility between subgenomes of resynthesized and natural *Brassica napus* using ATAC and ChIP-seq, and exploring the relationship between chromatin state, various epigenetic markers, and gene expression.

I found the work a valuable contribution to our knowledge of allopolyploid genome evolution and the comparison of diploid, synthetic, and natural polyploid is certainly unique and informative. The experimental approaches taken seemed to be broadly sound, though needing more details and slight corrections in some spaces, (additionally I am admittedly not very familiar with ATAC and ChIP-seq analysis), and the paper generally seemed well-argued.

My major concern with the paper relates to the presentation of much of the results. The number of figures and subfigures and their format was often an impediment to being able to understand and interpret the results. This is especially the case for figures 2 and 6, but many of the concerns apply to a lesser extent to other figures. I'd urge the authors to consider ways to change the way results are presented to help with legibility and clarity.

Major comments:

Figures

The core concern with many figures is too many subfigures and redundant information being plotted, and inconsistencies of coloring choices in plots that make interpretation difficult.

As an example, figure 2 contains several largely redundant subfigures because results are presented as a series of three pairwise comparisons instead of a single three-way comparison (as done in Fig 1b). By doing so the number of Venn diagrams would be reduced while maintaining the same amount of information, and the number of pie charts is cut in half by only needing to show RAC, NAC, and A_C specific, and common ACRs once. Additionally, I believe the pie chart graphics would be made easier to read by having a single, larger legend rather than smaller legends.

Likewise, Figure 3c includes several multi-comparison Venn diagrams where, by my understanding, several entries will always be equal to zero because there are mutually exclusive overlaps. These might be better presented as an Upset plot since zeroed entries can be excluded from the plot. It again may also be possible to slim the three paired comparisons to a single three-way comparison and plot results in those terms.

For figure 5, even as a pdf, the legends highlighting the colors of each rank are not legible, perhaps a single larger legend could be used instead.

Figure 6, inconsistent color patterns became an issue. In some panels, blue is RAC in others green is RAC, etc. As the legends are too small to easily read, this became an impediment to interpretation so using consistent color patterns and/or larger, more clear figure legends would benefit the manuscript.

Figure 8a it is unclear which plot corresponds to which genotype (in silico hybrid, synthetic polyploid, and natural polyploid).

RNA-seq

While generally, I believe the methodology to be sound, there are places that need elaboration or minor changes. For example, in the RNA-seq section, the authors say that reads were aligned to the reference genome with Pinfish, but from what I can gather from the github page, Pinfish does not perform alignments itself, it is a wrapper for several tools to make genome annotations based on long-

read RNA data. I think Pinfish might be calling MiniMap2 for its alignment. At the very least the authors should provide more detail on what tools are really being used and the settings for the alignment.

For the homoeolog expression bias analysis it would provide a better picture of the results if figure 7 plotted the log transformed TPM. It should also be stated if genes with TPM < 1 were filtered out, this would be recommended to eliminate genes that have low expression for artifactual reasons, as well as genes with tissue-specific expression patterns.

DNA Methylation

For the DNA methylation analysis, I would like the authors to provide more details about the library construction or analysis of the data so assess how they corrected for non-conversion rates of Uracil. Either a lambda spike-in should be used for this correction or analysis using the plastid genome should be used (see e.g. Lister et al. 2008 <https://doi.org/10.1016/j.cell.2008.03.029> and Bird et al. 2021 <https://doi.org/10.1111/nph.17137>) It's unclear as written if/how BatMeth2 accounts for non-conversion in ways that, for example, Methylpy does. Accounting for non-conversion of uracil however is a crucial quality control step for DNA methylation analysis.

Additionally, for methylation levels, the authors should state whether methylation levels were weighted by sequencing depth for a region, termed weighted methylation levels by Schultz et al. 2012 (<https://doi.org/10.1016/j.tig.2012.10.012>) as this is also best practice for analyzing DNA methylation data.

Minor comments:

While generally the literature review is thorough, the authors may want to cite and discuss some relevant work on subgenome asymmetry and epigenomic regulation of resynthesized *B. napus* like the small RNA analysis Martinez Palacios et al. 2019 (<https://doi.org/10.1093/molbev/msz007>) or the gene expression and DNA methylation results from Bird et al. 2021 (<https://doi.org/10.1111/nph.17137>), similarly the results from Renny-Byfield et al. 2017 (<https://doi.org/10.1093/molbev/msx121>) which looks at the relationship between biased fractionation, asymmetric homoeolog expression, and MNase sensitivity.

At line 141 the authors compare the RPKM (typo RPFM) between natural and resynthesized *B. napus* arguing for greater chromatin accessibility based on the difference in RPKM (700 vs 770) but how do the authors know this is a significant difference, how does this magnitude compare to the variation between replicates for example?

The results about cis/trans regulators is interesting, but the language is a little confusing. The authors say 'genotype-unique TFs' at line 270, but are the TFs actually unique to a genotype or is it just the accessibility of the motifs vary, and this produces genotype-unique TF binding?

Reviewer #2 (Remarks to the Author):

This manuscript tried to address the changes in chromatin accessibility and their relationship with epigenetic modifications and gene expression caused by polyploidization. The results provide valuable information on understanding how chromatin accessibility shapes transcriptome patterns during allopolyploidization and subsequent evolutionary processes.

Major comments

The results were not concisely written, some information in M&M did not need to repeat in results. The results are not related to the topic you want to address here can be shortened.

In discussion, authors focused the discussion on chromatin accessibility on different regions of genomes such as TSS, gACRs and iACRs, whereas no much discussion was given on how the

polyploidization and evolution shaped chromatin accessibility among these genotypes.

Minor comments

102: chromatin accessibility and its relationships with four epigenetic modifications, which four epigenetic modifications? Clarify here.

140: whereas resynthesized B. napus and natural B. napus had higher accessibility in these regions, with an average maximum of approximately 700 and 770 RPFM around TSS, add respectively to this sentence

266-0269: The statement was complicated making difficulty to understand. Rewrite.

Please carefully check the errors in the manuscript, such as tagmentation etc.

Reviewer #3 (Remarks to the Author):

This manuscript mainly described chromatin accessibility in natural allopolyploid B. napus, resynthesized B. napus, and in silico hybrid, and how it related to gene expression levels, histone modifications, and DNA methylations. In addition, this study investigated the difference of asymmetric chromatin accessibility between subgenomes of B. napus. In general, this comprehensive study expands understanding of relationships between chromatin accessibility and gene expression in allopolyploidization with B. napus as a model system.

Some weaknesses need to be addressed:

1. The analysis was mainly focused on regions around TSS, which is reasonable because usually accessible regions are highly enriched in promoter regions, however, it would be more logical to investigate the whole genome first, and then narrow down to the enriched part. Some previous studies indicate that, in addition to the promoter regions, intergenic regions also contribute to the chromatin accessibility, which is also shown in this study (About 30-40% ACRs are distributed within distal intergenic regions). However, downstream analysis did not investigate the intergenic regions.

2. The title is "Asymmetric subgenomic chromatin architecture impacts...", but analysis about asymmetric sub-genome is a relatively small proportion in the manuscript. Either name of the manuscript or the contents need to be adjusted.

To improve the asymmetric sub-genome chromatin accessibility part, I suggest adding a few analyses such as comparing the window based chromatin accessibility difference for the whole sub-genome between Cn and An (probably boxplots would be appropriate for the comparison). Contribution of different genomic regions (e.g. promoter, genic, intergenic, and etc.) needs to be shown for each sub-genome as well, due to the same reason as the above comment (intergenic regions could possibly also contribute a lot for the chromatin accessibility). In addition, I also suggest investigating how chromatin accessibility distributes along chromosomes for different sub-genomes.

3. Some parts in the paper are not easily readable to me, due to unclear logical links between observations and conclusions.

In addition to the above general suggestions, some specific comments are listed as follows:

1. Fig 1c, is "pACRs" a typo?

2. Fig 1e, pie chart is not a good visualization especially comparing across multiple pie charts. I suggest making a combined histogram with different colors showing A_C, RAC and NAC for comparison.

3. Line 172, is "13,399" a typo?

4. Figure 2a-c, same comment with figure 1e, pie char is not suitable for making comparisons. I suggest combining common and genotype specific ACRs of each genotype into single histogram.

5. Line 230, "Surprisingly, compared with the in silico 'hybrid', both up-regulated and down-regulated DEGs had higher ATAC-Seq signals in resynthesized and natural B. napus". However, the figure shows up-regulated DEGS in RAC is barely higher, almost same compared with A_C.

6. Line 245-253, "As shown in Fig. 3c, 24.9 % (69) of in silico 'hybrid'-enriched DEGs and 25.0% (90) of resynthesized B. napus-enriched DEGs were associated with genotype-enriched ACRs. In the

comparison of two types of *B. napus*, 27.0 % (176) of the resynthesized *B. napus*-enriched DEGs and 24.4% (162) of the natural *B. napus*-enriched DEGs were regulated by ACRs". These are relatively small proportions, how can it be concluded that "These results indicated that the changes in chromatin accessibility regulated the differential expression of genes".

7. Supple fig. 7, authors forgot to label a, b, c.

8. Line 300, "Surprisingly, the intensity of ACRs was positively associated not only with three histone modification levels (Fig. 5a-c), but also with the DNA methylation level", a correlation plot with R2 will be better evidence to make conclusions.

9. Figure 5, are "gOCR" and "iOCRs" typos?

10. Line 310-312, "highly expressed genes not only had the highest H3K4me3 and H3K27ac, but also had the highest H3K27me3 signal, followed by moderately expressed genes". It is true, but H3K27me3 as a repressive mark, it shows quite different profiles from the other two, what might be the reasons?

11. Line 321-322, "down-regulated DEGs showed higher H3K27me3 levels than up-regulated DEGs in all comparisons". Seems not true in NAV vs RAC.

12. Line 333-335, "differential gene expression seemed to be finely regulated by active histone modifications (H3K4me3 and H3K27ac) and DNA methylation in the three genotypes", I did not follow the logic link here, please rephrase with better explanations.

13. Figure 8a, what is each figure about? Please explain in legends.

14. Line 358, there is no figure 7f.

15. Line 373, "around TSS" needs to have a more specific interval range to avoid confusions.

16. Figure 9a, I would suggest removing outliers to get better visualization.

Besides, some minor issues need to be corrected. For example, there are some typos in the main text and figures. Figures need better visualization, e.g. legend/letter size to be adjusted to avoid being extra small.

Responses to the Reviewer 1's comments:

Comment 1: The core concern with many figures is too many subfigures and redundant information being plotted, and inconsistencies of coloring choices in plots that make interpretation difficult. As an example, figure 2 contains several largely redundant subfigures because results are presented as a series of three pairwise comparisons instead of a single three-way comparison (as done in Fig 1b). By doing so the number of Venn diagrams would be reduced while maintaining the same amount of information, and the number of pie charts is cut in half by only needing to show RAC, NAC, and A_C specific, and common ACRs once. Additionally, I believe the pie chart graphics would be made easier to read by having a single, larger legend rather than smaller legends.

Reply: Thank you for your valuable suggestion. In response, we have now used a single three-way comparison instead of previous three comparisons (see Fig 2a) and used a histogram instead of pie chart graphics (see Fig 2b). We also changed the colors or enlarged the legends in some figures to make them easier to read (such as Fig. 3).

Comment 2: Likewise, Figure 3c includes several multi-comparison Venn diagrams where, by my understanding, several entries will always be equal to zero because there are mutually exclusive overlaps. These might be better presented as an Upset plot since zeroed entries can be excluded from the plot. It again may also be possible

to slim the three paired comparisons to a single three-way comparison and plot results in those terms.

Reply: Thank you for your valuable suggestion. We have now replaced the previous Venn diagrams with a single Upset plot (see Figure 3c).

Comment 3: For figure 5, even as a pdf, the legends highlighting the colors of each rank are not legible, perhaps a single larger legend could be used instead.

Reply: Thank you for your valuable suggestion and careful checks. We have now enlarged the legend so that we can see it clearly (see Figure 5).

Comment 4: Figure 6, inconsistent color patterns became an issue. In some panels, blue is RAC in others green is RAC, etc. As the legends are too small to easily read, this became an impediment to interpretation so using consistent color patterns and/or larger, more clear figure legends would benefit the manuscript.

Reply: Thank you for your valuable suggestion and careful checks. We have now unified the colors and enlarged the legend to make the figure clearer (see Figure 6).

Comment 5: Figure 8a it is unclear which plot corresponds to which genotype (*in silico* hybrid, synthetic polyploid, and natural polyploid).

Reply: Thank you for your careful checks. We have now added the name of genotype in the plots corresponding to each genotype (see Figure 8a).

Comment 6: While generally, I believe the methodology to be sound, there are places that need elaboration or minor changes. For example, in the RNA-seq section, the authors say that reads were aligned to the reference genome with Pinfish, but from what I can gather from the github page, Pinfish does not perform alignments itself, it is a wrapper for several tools to make genome annotations based on long-read RNA data. I think Pinfish might be calling MiniMap2 for its alignment. At the very least the authors should provide more detail on what tools are really being used and the settings for the alignment.

Reply: Thank you for your valuable suggestions. We have now provided detail on reference genome alignment through Pinfish process (see line 580-585).

Comment 7: For the homoeolog expression bias analysis it would provide a better picture of the results if figure 7 plotted the log transformed TPM. It should also be stated if genes with $TPM < 1$ were filtered out, this would be recommended to eliminate genes that have low expression for artifactual reasons, as well as genes with tissue-specific expression patterns.

Reply: Thank you for your valuable suggestions. In fact, the log transformed TPM could change the original average value of the data (If the mean of X is greater than Y, but the variance of Y is larger, the mean of $\ln X$ after log transformation is less than $\ln Y$. it can be expressed by the formula, i.e, when $(X_1+X_2+\dots+X_n) > (Y_1+Y_2+\dots+Y_n)$, if $X_1 * X_2 * \dots * X_n < Y_1 * Y_2 * \dots * Y_n$, ($\ln X_1 + \ln X_2 + \dots + \ln X_n < \ln Y_1 + \ln Y_2 + \dots + \ln Y_n$)). So, we can't plot the log transformed TPM instead of the original data. But we removed

the value of TPM < 1 and redrew figure 7a (see figure 7a).

Comment 8: For the DNA methylation analysis, I would like the authors to provide more details about the library construction or analysis of the data so assess how they corrected for non-conversion rates of Uracil. Either a lambda spike-in should be used for this correction or analysis using the plastid genome should be used (see e.g. Lister et al. 2008 <https://doi.org/10.1016/j.cell.2008.03.029> and Bird et al. 2021 <https://doi.org/10.1111/nph.17137>). It's unclear as written if/how BatMeth2 accounts for non-conversion in ways that, for example, MethyIpy does. Accounting for non-conversion of uracil however is a crucial quality control step for DNA methylation analysis.

Reply: Thank you for your valuable suggestions. We have now provided detail on library construction and analysis of the data for DNA methylation (see line 606-621). A lambda spike-in should be used for corrected for non-conversion rates of uracil, we have the relevant details in the revised manuscript, as “The lambda DNA spike-in was used to correct for non-conversion rates of uracil by adding 1 ng of methyl-free lambda DNA to 1 µg genomic DNA as an internal reference for the conversion test.” (see line 607-610).

Comment 9: Additionally, for methylation levels, the authors should state whether methylation levels were weighted by sequencing depth for a region, termed weighted methylation levels by Schultz et al. 2012 (<https://doi.org/10.1016/j.tig.2012.10.012>) as

this is also best practice for analyzing DNA methylation data.

Reply: Thank you for your valuable suggestions. We have now stated that Batmeth2 calculates regional DNA methylation level by weighting the sequencing depth of a region, as “Batmeth2 calculates regional DNA methylation level by weighting the sequencing depth of a region” (see line 616-617).

Comment 10: While generally the literature review is thorough, the authors may want to cite and discuss some relevant work on subgenome asymmetry and epigenomic regulation of resynthesized *B. napus* like the small RNA analysis Martinez Palacios et al. 2019 (<https://doi.org/10.1093/molbev/msz007>) or the gene expression and DNA methylation results from Bird et al. 2021 (<https://doi.org/10.1111/nph.17137>) , similarly the results from Renny-Byfield et al. 2017 (<https://doi.org/10.1093/molbev/msx121>) which looks at the relationship between biased fractionation, asymmetric homoeolog expression, and MNase sensitivity.

Reply: Thank you for your valuable suggestions. We have now cite and discuss these literatures in our revised manuscript, as “ Newly formed allopolyploids need to reorganize distinct genomes from different parental species, which disrupts genetic and epigenomic featuresand results in altered DNA methylation, histone markers, chromatin compactness, sRNA production, and gene expression” and “In resynthesized *B. napus*, subgenome dominance is partly related to DNA methylation difference between two subgenomes over the first ten generations postpolyploidisation. In maize, biased fractionation between two ancient subgenomes

not only via DNA methylation, but also in MNase sensitivity” (see line 415; line 513-516).

Comment 11: At line 141 the authors compare the RPKM (typo RPFM) between natural and resynthesized *B. napus* arguing for greater chromatin accessibility based on the difference in RPKM (700 vs 770) but how do the authors know this is a significant difference, how does this magnitude compare to the variation between replicates for example?

Reply: Thank you for your valuable suggestion and careful checks. We have now corrected the typos in the revisions (see line 141). T test was used to test the statistical significance of the differences and chromatin accessibility of natural *B. napus* is indeed higher than that of resynthesized *B. napus*. We have now illustrated the significant differences in figure legend, as “e Profiles of ACRs. T test revealed statistical significance of the differences of chromatin accessibility among the three genotypes.” (see line 846-847).

Comment 12: The results about cis/trans regulators is interesting, but the language is a little confusing. The authors say ‘genotype-unique TFs’ at line 270, but are the TFs actually unique to a genotype or is it just the accessibility of the motifs vary, and this produces genotype-unique TF binding?

Reply: Thank you for your valuable suggestion and careful checks. We have now corrected the previously confusing language, as “These results implied that many

genotype-unique over-represented motifs were bound by corresponding TFs to regulate the differential expression of genes.” (see line 256-258).

We would like to thank Reviewer 1 for the constructive and useful suggestions. It is very helpful for revising and improving our manuscript. We have answered the questions one by one and revised the manuscript.

Responses to the Reviewer 2’s comments:

Comment 1: The results were not concisely written, some information in M&M did not need to repeat in results. The results are not related to the topic you want to address here can be shortened.

Reply: Thank you for your valuable suggestion. We have now deleted some repeated information in results (see line 136; line 229-231; line 258 in original manuscript). We cut out some results irrelevant to the subject under discussion (see line 183-195 in original manuscript).

Comment 2: In discussion, authors focused the discussion on chromatin accessibility on different regions of genomes such as TSS, gACRs and iACRs, whereas no much discussion was given on how the polyploidization and evolution shaped chromatin accessibility among these genotypes.

Reply: Thank you for your valuable suggestion. We have now added a discussion of how polyploidy and evolution shaped chromatin accessibility among these genotypes,

as “Surprisingly, the ATAC-Seq signal of ACRs in resynthesized *B. napus* was significantly higher than that in in silico ‘hybrid’ but was significantly lower than that in natural *B. napus*, which indicated that both allopolyploidization and subsequent evolution increased chromatin accessibility of *B. napus*.” (see line 435-438).

Comment 3: 102: chromatin accessibility and its relationships with four epigenetic modifications, which four epigenetic modifications? Clarify here.

Reply: Thank you for your valuable suggestion and careful checks. We have now clarified four epigenetic modification specifically three histone epigenetic modifications (H3K4me3, H3K27ac and H3K27me3) and DNA methylation (see line 102-103).

Comment 4: 140: whereas resynthesized *B. napus* and natural *B. napus* had higher accessibility in these regions, with an average maximum of approximately 700 and 770 RPKM around TSS, add respectively to this sentence.

Reply: Thank you for your valuable suggestion. We have now added respectively to the sentence (see line 145).

Comment 5: 266-0269: The statement was complicated making difficulty to understand. Rewrite.

Reply: Thank you for your valuable suggestion. We have now rewritten this sentence as “We overlapped these target genes and DEGs, and found 37.5%, 34.6%, and 35.6%

DEGs were regulated by these TFs in comparisons of three genotypes, respectively (Supplementary Fig. 6e). These results implied that many genotype-unique over-represented motifs were bound by corresponding TFs to regulate the differential expression of genes.” (see line 254-258).

Comment 6: Please carefully check the errors in the manuscript, such as tagmentation etc.

Reply: Thank you for your valuable suggestion and careful checks. We have now corrected the typos we checked, such as “tagmentation” in line 552 and “yje” in line 324 in original manuscript.

We would like to thank Reviewer 2 for the constructive and useful suggestions. It is very helpful for revising and improving our manuscript. We have answered the questions one by one and revised the manuscript.

Responses to the Reviewer 3’s comments:

Comment 1: The analysis was mainly focused on regions around TSS, which is reasonable because usually accessible regions are highly enriched in promoter regions, however, it would be more logical to investigate the whole genome first, and then narrow down to the enriched part. Some previous studies indicate that, in addition to the promoter regions, intergenic regions also contribute to the chromatin accessibility, which is also shown in this study (About 30-40% ACRs are distributed within distal

intergenic regions). However, downstream analysis did not investigate the intergenic regions.

Reply: Thank you for your valuable suggestion. We have now added the genome-wide chromatin accessibility analysis (see Supplementary Fig. 3) and intergenic regions chromatin accessibility analysis (see Fig. 7).

Comment 2: The title is “Asymmetric subgenomic chromatin architecture impacts....”, but analysis about asymmetric sub-genome is a relatively small proportion in the manuscript. Either name of the manuscript or the contents need to be adjusted. To improve the asymmetric sub-genome chromatin accessibility part, I suggest adding a few analyses such as comparing the window based chromatin accessibility difference for the whole sub-genome between C_n and A_n (probably boxplots would be appropriate for the comparison). Contribution of different genomic regions (e.g. promoter, genic, intergenic, and etc.) needs to be shown for each sub-genome as well, due to the same reason as the above comment (intergenic regions could possibly also contribute a lot for the chromatin accessibility). In addition, I also suggest investigating how chromatin accessibility distributes along chromosomes for different sub-genomes.

Reply: Thank you for your valuable suggestion. We have now added few analyses such as chromatin accessibility difference for the whole sub-genome between C_n and A_n (see Fig. 7b) and investigated the contribution of different genomic regions (intergenic, exon, intron, promoter-TSS and TTS, see Fig. 7c). Chromatin

accessibility distributes along chromosomes can be seen in Supplementary Fig. 3.

Comment 3: Some parts in the paper are not easily readable to me, due to unclear logical links between observations and conclusions.

Reply: Thank you for your valuable suggestion. We have now rewritten some inaccurate conclusions, such as line 242; line 256-258; line 308-309; line 319-321 and so on.

Comment 4: Fig 1c, is “pACRs” a typo?

Reply: Thank you for your careful checks. We have now corrected the typo as “iACRs”(see Fig 1c).

Comment 5: Fig 1e, pie chart is not a good visualization especially comparing across multiple pie charts. I suggest making a combined histogram with different colors showing A_C, RAC and NAC for comparison.

Reply: Thank you for your valuable suggestion. We have now replaced the pie charts with a combined histogram (see Fig 1f).

Comment 6: Line 172, is “13,399” a typo?

Reply: Thank you for your careful checks. We have now corrected the typo (see line 173-176).

Comment 7: Figure 2a-c, same comment with figure 1e, pie chart is not suitable for making comparisons. I suggest combining common and genotype specific ACRs of each genotype into single histogram.

Reply: Thank you for your valuable suggestion. We have now used a single venn diagram instead of three venn diagrams in Figure 2a-c (see Figure 2a) and replaced the pie charts with a combined histogram (see Figure 2b). We also corrected the content in Figure 2d (see Figure 2c) the corresponding content in the manuscript (see line 173-176; line 181-185).

Comment 8: Line 230, “Surprisingly, compared with the in silico ‘hybrid’, both up-regulated and down-regulated DEGs had higher ATAC-Seq signals in resynthesized and natural *B. napus*”. However, the figure shows up-regulated DEGS in RAC is barely higher, almost same compared with A_C.

Reply: Thank you for your careful checks. In fact, the ATAC-Seq signals of up-regulated DEGs of RAC vs A_C in RAC and A_C looked very close due to the large ordinate value, but there are significant differences between them (T test was used to test the statistical significance). When the ordinate value is reduced, the distance between the ATAC-Seq signals of up-regulated DEGs in RAC and A_C will become further. We have now redrawn the figure (see figure 3b).

Comment 9: Line 245-253, “As shown in Fig. 3c, 24.9 % (69) of in silico ‘hybrid’-enriched DEGs and 25.0% (90) of resynthesized *B. napus*-enriched DEGs

were associated with genotype-enriched ACRs. In the comparison of two types of *B. napus*, 27.0 % (176) of the resynthesized *B. napus*-enriched DEGs and 24.4% (162) of the natural *B. napus*-enriched DEGs were regulated by ACRs”. These are relatively small proportions, how can it be concluded that “These results indicated that the changes in chromatin accessibility regulated the differential expression of genes”.

Reply: Thank you for your careful checks. We have now corrected the inaccurate conclusion as “These results indicated that the changes in chromatin accessibility regulated the differential expression of a part of DEGs, which was similar in maize” (see line 242).

Comment 10: Supple fig. 7, authors forgot to label a, b, c.

Reply: Thank you for your careful checks. We have now added label a, b and c in Supple Fig. 7 (see Supplementary Fig. 7)

Comment 11: Line 300, “Surprisingly, the intensity of ACRs was positively associated not only with three histone modification levels (Fig. 5a-c), but also with the DNA methylation level”, a correlation plot with R2 will be better evidence to make conclusions.

Reply: Thank you for your valuable suggestion. We have now added the correlation analysis and we've come to the same conclusion (see Supplementary Fig. 10).

Comment 12: Figure 5, are “gOCR” and “iOCRs” typos?

Reply: Thank you for your careful checks. We have now corrected the typos as “gACRs” and “iACRs”(see Figure 5d-f).

Comment 13: Line 310-312, “highly expressed genes not only had the highest H3K4me3 and H3K27ac, but also had the highest H3K27me3 signal, followed by moderately expressed genes”. It is true, but H3K27me3 as a repressive mark, it shows quite different profiles from the other two, what might be the reasons?

Reply: Thank you for your careful checks. H3K27me3 has long been considered as an repressive histone modification. In plants, H3K27me3 is required for normal expression of development-import genes and it is mainly distributed in the transcription region of genes^{1,2}. H3K4me3 and H3K27ac, two active histone modifications, are mainly enriched in the TSS of genes^{2,3}. So, we can observe that the different distribution of H3K27me3 from the other two, which may be due to the different mechanism of the corresponding chromatin modifying enzyme deposition modification^{4,5}.

Comment 14: Line 321-322, “down-regulated DEGs showed higher H3K27me3 levels than up-regulated DEGs in all comparisons”. Seems not true in NAC vs RAC.

Reply: Thank you for your careful checks. We have now corrected the results as “Similar to the ATAC-Seq signals, down-regulated DEGs showed higher H3K27me3 levels than up-regulated DEGs except in comparison of resynthesized *B. napus* and natural *B. napus*” (see line 308-309).

Comment 15: Line 333-335, “differential gene expression seemed to be finely regulated by active histone modifications (H3K4me3 and H3K27ac) and DNA methylation in the three genotypes”, I did not follow the logic link here, please rephrase with better explanations.

Reply: Thank you for your careful checks and valuable suggestion. We have now rephrased this sentence as “The up-regulation of DEGs seemed to require active histone modifications (H3K4me3 and H3K27ac), but lower DNA methylation levels in the three genotypes.” (see line 319-321).

Comment 16: Figure 8a, what is each figure about? Please explain in legends.

Reply: Thank you for your careful checks. We have now added the name of genotype in the plots corresponding to each genotype (see Figure 8a).

Comment 17: Line 358, there is no figure 7f.

Reply: Thank you for your careful checks. We have now added the missing figure (see Supplementary Fig. 15) and corrected the name of figure (see line 358).

Comment 18: Line 373, “around TSS” needs to have a more specific interval range to avoid confusions.

Reply: Thank you for your valuable suggestion. We have now added the specific interval range in the manuscript, as “between -100 bp to 100 bp on either hand of TSS”

(see line 368-370). and “between -200 bp to 700 bp on either hand of TSS” (see line374).

Comment 19: Figure 9a, I would suggest removing outliers to get better visualization.

Reply: Thank you for your valuable suggestion. We have now removed outliers in Figure 9a.

Comment 20: Besides, some minor issues need to be corrected. For example, there are some typos in the main text and figures. Figures need better visualization, e.g. legend/letter size to be adjusted to avoid being extra small.

Reply: Thank you for your valuable suggestion. We have now corrected the typos we checked and enlarged the legend/letter appropriately.

We would like to thank Reviewer 3 for the constructive and useful suggestions. It is very helpful for revising and improving our manuscript. We have answered the questions one by one and revised the manuscript.

References:

1. Zhang, X. et al. Whole-genome analysis of histone H3 lysine 27 trimethylation in *Arabidopsis*. *PLoS Biol.* **5**, e129 (2007).
2. Zhang, A., et al. Profiling of H3K4me3 and H3K27me3 and their roles in gene subfunctionalization in allotetraploid cotton. *Front Plant Sci.* **12**,761059 (2021).
3. Ricci, W.A. et al. Widespread long-range cis-regulatory elements in the maize

genome. *Nat. Plants* **5**, 1237–1249 (2019).

4. Gan, E.S., Xu, Y. & Ito, T. Dynamics of H3K27me3 methylation and demethylation in plant development. *Plant Signal Behav.* **10**, e1027851 (2015).
5. Beacon, T.H., Delcuve, G.P., López, C., Nardocci, G., Kovalchuk, I., van Wijnen, A.J., Davie, J.R. The dynamic broad epigenetic (H3K4me3, H3K27ac) domain as a mark of essential genes. *Clin Epigenetics.* **13**, 138 (2021).

Reviewer #1 (Remarks to the Author):

This revised version has sufficiently addressed all prior concerns and I have no further comments.

Reviewer #2 (Remarks to the Author):

Final correction such as grammar etc is needed, for example:

We overlapped these target genes and DEGs, and found 37.5%, 34.6%, and 35.6% DEGs were regulated by these TFs in comparisons of three genotypes, respectively (Supplementary Fig. 6e)
Unclear statement, which value for which genotype?

Line 368-375: "on either hand of TSS"?

"In maize, biased fractionation between two ancient subgenomes not only via DNA methylation, but also in MNase sensitivity", not good sentence

Reviewer #3 (Remarks to the Author):

Thank you for making efforts to revise the manuscript.

Majority of my questions have been clearly addressed, except for the part of intergenic regions. The authors added some analysis such as fig. 7c and Supple fig. 3, but did not provide detailed explanation about the results or discussions, for example, in fig. 7c, intensity of ACRs is much higher than any other regions even promoter, and surprisingly the comparison results were opposite than the rest for A_C, with An genome showing higher intensity than Cn, what might be reasons. Besides, due to important role of ACRs in distal intergenic regions in evolutionary process as mentioned in the manuscript, I was expecting some deeper exploration about intergenic regions, such as, enrichment of histone modifications in intergenic regions, and etc., which were not shown in the revised paper.

Another minor comment, line 181, authors mentioned "More common ACRs were distributed in the distal intergenic region but less of that distributed in the promoter region than genotype-specific ACRs." Is there any number, statistic or figures to support the difference?

Responses to the Reviewer 1's comments:

This revised version has sufficiently addressed all prior concerns and I have no further comments.

We would like to thank Reviewer 1 for the positive comments.

Responses to the Reviewer 2's comments:

Comment 1: Final correction such as grammar etc is needed, for example: We overlapped these target genes and DEGs, and found 37.5%, 34.6%, and 35.6% DEGs were regulated by these TFs in comparisons of three genotypes, respectively (Supplementary Fig. 6e). Unclear statement, which value for which genotype?

Reply: Thank you for your valuable suggestion. In response, we have now revised the unclear statement, as "We overlapped these target genes and DEGs, and found 37.5%, 34.6%, and 35.6% DEGs were regulated by these TFs in comparisons of *in silico* 'hybrid' vs. resynthesized *B. napus*, *in silico* 'hybrid' vs. natural *B. napus* and resynthesized *B. napus* vs. natural *B. napus*, respectively (Supplementary Fig. 6e)" (see lines 257-258).

Comment 2: Line 368-375: "on either hand of TSS"?

Reply: Thank you for your careful check. In response, we have now revised the unclear statement (see lines 374-380).

Comment 3: "In maize, biased fractionation between two ancient subgenomes not only

via DNA methylation, but also in MNase sensitivity", not good sentence.

Reply: Thank you for your careful check. In response, we have now rewritten this sentence, as "In maize, DNA methylation and chromatin accessibility drive biased fractionation between two ancient subgenomes" (see lines 521-522).

We would like to thank Reviewer 2 for the constructive and useful suggestions. It is very helpful for revising and improving our manuscript. We have answered the questions one by one and revised the manuscript.

Responses to the Reviewer 3's comments:

Comment 1: Majority of my questions have been clearly addressed, except for the part of intergenic regions. The authors added some analysis such as fig. 7c and Supple fig. 3, but did not provide detailed explanation about the results or discussions, for example, in fig. 7c, intensity of ACRs is much higher than any other regions even promoter, and surprisingly the comparison results were opposite than the rest for A_C, with An genome showing higher intensity than Cn, what might be reasons. Besides, due to important role of ACRs in distal intergenic regions in evolutionary process as mentioned in the manuscript, I was expecting some deeper exploration about intergenic regions, such as, enrichment of histone modifications in intergenic regions, and etc., which were not shown in the revised paper.

Reply: Thank you for your valuable suggestion. We have now added the analysis about enrichment of histone modifications and DNA methylation in intergenic regions (see

lines 344-346, Supple fig. 16 and 17).

Comment 2: Another minor comment, line 181, authors mentioned “More common ACRs were distributed in the distal intergenic region but less of that distributed in the promoter region than genotype-specific ACRs.” Is there any number, statistic or figures to support the difference?

Reply: Thank you for your valuable suggestion. Figure 2b supported the difference and we have now added the number to support the difference, as “More common ACRs were distributed in the distal intergenic region (5.6% -12.4%) but less of that distributed in the promoter region (4.8%-11.2%) than genotype-specific ACRs (Fig. 2b)” (see lines 182-183).

We would like to thank Reviewer 3 for the constructive and useful suggestions. It is very helpful for revising and improving our manuscript. We have answered the questions one by one and revised the manuscript.